# *Earth Girl Volcano*: Characterizing and Conveying Volcanic Hazard Complexity in an Interactive Casual Game of Disaster Preparedness and Response

Isaac Kerlow[1, 2], Gabriela Pedreros[3], Helena Albert[4, 5]

[1] art-science-media.com, Los Angeles, United States of America
[2] Earth Observatory of Singapore, Nanyang Technological University, Singapore
[3] Observatorio Volcanológico de Los Andes del Sur, SERNAGEOMIN, Temuco, Chile
[4] Central Geophysical Observatory, Spanish Geographic Institute (IGN), Madrid, Spain
[5] Departament de Mineralogia, Petrologia i Geologia Aplicada, University of Barcelona, Barcelona, Spain

*Correspondence to*: Isaac Kerlow (ikerlow@gmail.com)

**Abstract.** This paper focuses on the process of developing the *Earth Girl Volcano* game, and presents some of our best
professional practices and lessons-learned. The paper shares our experience of weaving storytelling in the not-so-straightforward process of interdisciplinary collaboration between artists and scientists. Our practice-based research approach to games is centered around a diligent and rigorous game development method that is story-centric and that uses storytelling to communicate scientific concepts. Our development methodology is presented in detail without the usual focus on quantitative evaluations: games are not scientific projects but audiovisual interactive catalysts of engagement. The
survival of many communities during volcanic emergencies is tied to their knowledge of volcanic preparedness. Unfortunately, there is a gap between scientific terminology and the non-technical language used by the general population. For this reason it is necessary to develop and implement engaging outreach strategies that familiarize communities at risk with volcanic hazards, that show how a volcanic event unfolds and what to do in case of an emergency. Interactive games provide a perfect alternative to engage communities and to impact their resilience. The *Earth Girl Volcano* game is about
making strategic decisions that minimize risk in communities exposed to volcanic hazards. Conveying the complexities of volcano disaster preparedness to a general audience is a communication challenge in itself, because of the multi-layered, interrelated and technical nature of the information. We use interactive dramatizations of hazard scenarios with people for players to identify with the characters in the game and to empathize with communities impacted by volcanic hazards. We present our approach for characterizing multiple hazard scenarios and dynamics in precise but nimble ways, and for
designing engaging gameplay within the context of a casual strategy game. We seek to engage mainstream audiences and familiarize them with volcanic evacuations and disaster risk management by providing a high degree of playability, using storytelling to create empathy, making creative use of staging and visuals, and using plain language. We believe that the combination of all these techniques yields a whole that is greater than the sum of its parts, a perfect storm that is able to create an emotional connection between players and the hazard scenarios presented in the game.

## 35  1 Introduction

Volcanic eruptions are a natural hazard that manifests itself in over 85 countries throughout the world. It is estimated that 800 million people live within a 100 km radius of active volcanic systems and are exposed to volcanic risk (Loughlin, 2015). The initial motivation behind our work was to develop and produce free-to-download and fun-to-play games of natural hazards preparedness and response. We were also interested in creating games that facilitate learning, games based on
scientific knowledge that provide practical knowledge for players to absorb while playing. Volcanology is a complex science with multiple dynamic variables, and understanding volcanic risks requires a significant amount of information including the eruptive history of a volcano and the geography of the area (Sparks, 2004), and the situation of the nearby population at risk.

Generally speaking there is limited knowledge about volcanic activity throughout the world and in many instances volcanic risks have not yet been evaluated (Lockwood, 2013). In light of these volcanic complexities we realized that any effective game on volcanic disaster preparedness and response would have to be simple and flexible, and easy to understand in a variety of cultural contexts.

A few of the critical team members had backgrounds in the professional production of videogames and computer games and we were keen on the idea of following the best-practices found in the commercial world of interactive game entertainment to create games that were inspired by scientific principles. We realized at the time that this represented an experimental approach that could only be successful through a partnership between scientists and game artists. We were intrigued by the possibilities and were lucky to have the opportunity to develop and explore our ideas.

The overall goal was to create interactive games that would be easy-to-play and easy-to-learn from, games that could contribute to the human preparedness initiative at-large. There are millions of humans living under the shadow of volcanic risk. According to the ranking by the National Geology and Mining Service (SERNAGEOMIN) of risk specific to active volcanoes in Chile, for example, around 1,500,000 people are exposed to some type of volcanic hazard within the Chilean territory. This figure represents about 8.3% of the total population and it includes individuals living in high to medium danger zones or in areas that are exposed to ash fall (Pedreros, 2020).

We were hopeful that given the proper distribution a robust series of natural hazards games could have a significant impact on global hazard awareness and preparedness. Ultimately the game seeks to empower communities in hazardous areas by improving their abilities of preparedness and response. For this purpose we met and talked directly to survivors of natural and human-made disasters and learned from those interactions a lot of factual experiential information (Kerlow, 2016). Based on the testimony of many survivors it seemed that a large majority of the casualties occur due to a lack of practical disaster preparedness knowledge, due to an inability to recognize the early signs of potential danger and the failure to act decisively. We got to work and in 2011 the early team released three simple interactive games with a focus on floods, tsunamis and volcanoes (Kerlow, 2011). We learned a lot from this early experience about features that are generally successful in disaster games for mainstream audiences. These early games require the Flash plug-in and they are still available at http://earthgirlgame.com/. In 2015 we released a more involved game focused exclusively on tsunamis, and in 2018 we released a volcano game. At time of writing the best two websites to download the English computer version of the game for Windows and MacOS are https://earthobservatory.sg/ and https://art-science-media.com/. Several localizations of the game for digital tablets are available at Google Play and the App Store.

Our game design and development methodology is framed by a few considerations and general techniques that are detailed below: using show and tell, considering the target audiences, optimizing for the chosen technology platforms, committing to a particular game format, using prototypes to develop the concept and the functionality of the game, apply rigorous quality control and user testing all throughout the process, and take advantage of interdisciplinary collaborations. The specific game design techniques used to characterize volcanic hazard complexity are detailed in Section 5.

**1.1 Show and Tell**

Watching the force of nature can be highly engaging, and what better way to create interest in volcanic hazards than to show volcanoes in action. The majestic and unstoppable quality of volcanoes instantly captivates people regardless of their background. For this reason we use volcanic activity as the main driver of the action in *Earth Girl Volcano*. Showing how the volcanic hazards develop and impact the environment allows players to connect the dots and literally see how different aspects of a single episode are connected. Representing on-screen the progress of the multiple hazards facilitates the understanding of volcanic episodes as systems of interrelated events, and facilitates a seamless visualization of precursor events. Showing the different impacts after an episode of volcanic activity allows players to better understand the difference between light damage and catastrophic damage, it also helps to understand what goes into preventing and minimizing catastrophic damage. At every point we try to convey the story by dramatizing and not by explaining. Increased realism in

interactive games has a direct effect on the players' attention and retention (Krcmar, 2010), and we use every technique available within our budget to maximize the realistic feeling of the volcanic hazards in *Earth Girl Volcano*. These techniques include 2D and 3D animation (Fig. 1), sound effects and music. The process of developing and producing a suitable visual style for this game was an interesting one but it is beyond the scope of this paper.

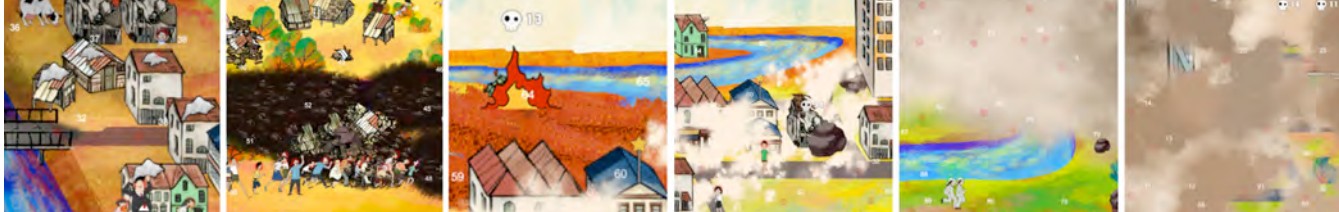

**Figure 1: Details of visual characterizations of volcanic hazards in the game (L to R): ash fall, mudflow, lava, gas and rock fall, partial and total burning clouds.**

A picture is worth a thousand words. Producers of TV News programs know that showing a snapshot of a natural disaster impacting people's lives creates instant strong emotions (Nazari, 2011; Lin, 2013). In *Earth Girl Volcano* we placed a major effort to represent unravelling volcanic hazards in impactful ways that are also grounded on real behaviors. To accomplish this result we used an innovative combination of 2D and 3D animation techniques that were visually rich but technically simple. This made possible the viewing of the animated simulations on devices and computers that are not state of the art.

**1.2 Target Audiences**

A major practical objective when the core Earth Girl team set out to develop natural hazard games in 2010 was to create games for an audience that was both mainstream and non-technical. By mainstream we understood an audience that might be vaguely aware of volcanic disasters but is largely unfamiliar and lacks expert knowledge. By non-technical we understood an audience that was not interested in learning about the technicalities of volcanology, Earth science or disaster risk management, including details such as monitoring gas emissions or reading preparedness lessons. We had played several of the hazard games available at the time and they were either too lesson-oriented or included too many technical terms to be of significant interest to a mainstream non-technical audience (Latawiec, 2019; Castronovo, 2017; Solinska-Nowak, 2018). Our objectives were sharpened as we took the time over a period of several years (2009-2017) to talk to survivors of natural hazards, civil defense personnel and volcano monitoring scientists in different countries throughout the world including Indonesia, The Philippines, Thailand, Vanuatu, China, Italy, Chile, Mexico and the United States. The lessons we learned from those conversations eventually led us to focus on two audience subsets that emerged as critical audiences: children and leaders, including government officials and community leaders, particularly those in disaster high-risk communities.

Generally speaking children are usually at high risk during a natural hazard because most of them rely on the adults to save them or to tell them what to do. But as it turns out there are many occasions before or during a natural hazard that for whatever reason reliable adults are not around and children must fend for themselves. In fact the genesis of the Earth Girl character pays homage to the many girls throughout South and Southeast Asia who tragically perished during the 2004 Indian Ocean tsunami mostly because they lacked proper disaster awareness and preparedness information. During the last decade we gave countless games demos to students in schools and universities because children represent a direct link to parents, siblings and older members of the family. In Chile, for example, the National Network of Volcanic Surveillance organizes outreach fairs throughout areas of high volcanic risk. The participation in the 2019 regional event in the Lago Ranco commune, Los Ríos region, included almost a thousand children (998) with 51.4% girls and 48.6% boys. Around 300 adults attended with 67% women and 37% men. The attendance to geology and volcanology workshops was 68% female and 32% male, while the emergency management workshop was 20% female and 80% male (Gho, 2019).

During the early phase of research for this project we heard from NGO staff and field volcanologists how challenging it is to convey details and implications of volcanic hazards to non-experts, especially during critical episodes (Jong, 2018; McBride, 2019). Presenting a holistic view of an emergency evacuation is challenging because multiple processes are connected and

125 happen in parallel, and because some of the technical and scientific concepts are complex. We have witnessed with satisfaction how the scenarios in the *Earth Girl Volcano* game have been used to illustrate and explain volcanic disaster preparedness, mitigation and response to non-specialists. The game provides a wide variety of scenarios that facilitates relating gameplay to specific hazard realities of diverse locations.

From the start we sought to make an interactive game with characteristics that would "speak to" or connect with non-
130 technical individuals who lived in or near regions with significant risk of natural hazards. A game they would want to play, a game that they would connect with and recommend to others (Fullerton, 2019). Many of the areas at risk are located in developing countries (Oppenheimer, 1998) and we assumed from the start that delivering games in relevant languages would be critical to their potential success. In the case of the *Earth Girl Volcano* game, for example, we identified the languages spoken throughout the Ring of Fire as a good collection of initial languages to maximize potential outreach impact. The
135 initial target languages for the first round of development included English, Spanish, Indonesian, Tagalog (The Philippines), Bislama and French (Vanuatu).

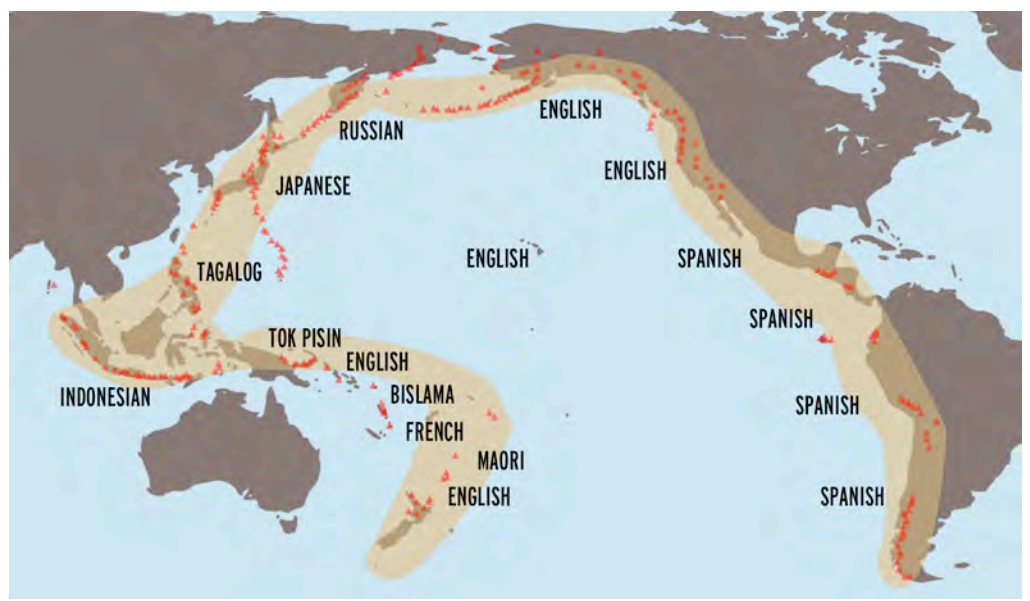

**Figure 2: Map showing some of the major official languages spoken around the Ring of Fire.**

We realized that generally speaking our potential audience would be multi-cultural and we had to be sensitive to the different
traditions and attitudes to volcanic preparedness and response (Donovan, 2014, 2019). We also knew from field experience (Kerlow, 2012) that our target audience typically had different degrees of preparedness awareness. Some residents of high-risk areas have considerable knowledge and experience in dealing with natural hazards. But many in the same locations have limited or no information about preparedness or response, let alone the basic scientific information that explains the hazards themselves. In either case we felt that a game that different types of non-experts would want to play could benefit both of
these populations at risk by increasing their practical knowledge through gameplay, particularly knowledge about early warning and disaster risk management. The plan was to provide learning by experiencing the *before* and *during* of a natural hazard, by virtually "being there," by virtually "participating." In summary the simple and straightforward gameplay of *Earth Girl Volcano* is meant to engage a target audience of pre-teens, early teens and adults. The ESRB (Entertainment Software Rating Board) rating is for EVERYONE.

**1.3 Technology Platforms**

In terms of the technology used to create and deploy the game our underlying strategy was to always keep it simple and stay away from solutions that would require high-end hardware to play the game. We also considered that our core target audiences had limited access to high-end computers or the latest gaming platforms, and a high-technology specification would have surely placed the games out of the reach of many of the target communities in the developing world. We

released *Earth Girl Volcano* as a downloadable for digital tablets and computers. We chose not to release the app for phones because based on early testing it was clear that a large percentage of test users had difficulty playing the game on smart phones with small screens. The play area was just too small and a massive redesign would have been necessary to make the experience possible, unfortunately we had neither the resources nor the opportunity to do this.

As mentioned above, the first three Earth Girl games can still be played online and offline on any device and browser that supports the Flash plug-in. These early games were also available on CD giveaway packs with stickers and posters that we distributed when demonstrating the game during Geography lessons at host elementary and middle schools. We are considering solutions to extend the lifetime of the first three games after 2020 when support for the Flash online plug-in is scheduled to end. One alternative that we have considered to extend these Flash games is to upload them to the repository website https://itch.io/. The second round of Earth Girl games, tsunami and volcano, were created in the Unity environment with each available as a standalone app. Both games run on iOS and Android tablets, and the volcano game is also available for Windows and MacOS computers. The multiple versions of the game are available online as free downloads, from Google Play, the App Store and a variety of public websites.

**1.4 Game Format - Why a Casual Game?**

*Earth Girl Volcano* combines features from a few game genres. It is primarily a casual game because each volcanic environment, or level, in the game can be played in just a few minutes. As with any casual game it is not necessary in *Earth Girl Volcano* to play all the locations in the game to benefit from it, players can enjoy the experience and learn by playing just one or a few levels. In this sense casual games, because they are light and not intimidating, can be effective vehicles to connect with communities at risk and decision makers. Early in the development process a number of the Earth scientists involved expressed concern about creating something "casual" in the context of a science-inspired game. The team realized at this point that a common understanding of terminology and goals would be necessary for the successful completion of the project. *Earth Girl Volcano* also has major elements of a traditional strategy game because players must devise a strategy and put it into action in order to win or survive. The game also has minor elements of adventure games and management games because players encounter multiple surprises and unknowns, as in a typical adventure game, and must manage resources in the most efficient possible way. The *Earth Girl Volcano* interactive game is about exploring a location, gathering information, coming up with a hazard defense and combat strategy, and putting that strategy into action in just about fifteen minutes.

There are many genres and formats of gaming, and we explored different possibilities in searching for the best way to structure and deliver the *Earth Girl Volcano* disaster game. Genres in gaming are like genres in Literature, each is best suited to different styles of storytelling and different types of stories. Once the design of a game is committed to a particular game genre it is important to take advantage and maximize the peculiarities of the genre in question. Consider the differences between a poem, a short story, an essay, and a novel; each has particular strengths and weaknesses depending on the topic at hand. When choosing a game format we considered practical issues such as our own resources and our target audiences. We had to find a concise and fun mechanism to deliver complex information, and a compact game format could be useful in forcing us to find that mechanism or face sure failure. There is nowhere to hide in a small game: it either works or it doesn't. It was also clear in the early stages that strategy was a promising tool to facilitate the non-technical players to "get into" the scientific and technical aspects of volcanic hazards. In the context of putting together a strategy for a volcanic hazard evacuation, science is just one of the critical elements but not the only one. So players unfamiliar or uncomfortable with science would quickly think of the game as an easy-to-play strategy game instead of an intimidating, long, or tedious science game (Weizman, 2014). It seemed that practical knowledge was likely to be more effectively absorbed by players in the context of a casual strategy game than with a long-form traditional strategy game or a science-focused simulation game.

**1.5 The Game Prototype**

Our overall approach to making disaster games is modelled on the creative cycle often used to develop and producing commercial independent games. Such workflow considers at the start of the process the general characteristics of the game and moves to build an early game prototype. These general characteristics include, for example, the target audience, the main objectives, the gaming style, the basic gameplay and game mechanics, the user interface, the visual look, and the basic technical specs (Kerlow, 2009). Game prototypes are bare-bones versions of the game with some functionality and limited interactivity, and are used widely throughout the industry (Schell, 2019). Prototypes are meant to show how specific ideas could be implemented into specific visuals and mechanisms. Prototypes are an approximation of the game as it grows and is slowly fine-tuned form a rough concept to a system of engaging situations and mechanisms. The multiple versions of the game prototype were our main vehicle to assemble, visualize and evaluate the different elements of the game as we made progress. The early proof of concept prototypes were useful to demonstrate to the dozen or so scientific collaborators that it was possible to make a game such as the one we envisioned. These early game prototypes were also extremely useful for the creative team to test ideas and quickly see and evaluate the results. Most importantly, the subsequent "builds" or iterations of the prototype became our primary vehicle of interdisciplinary collaboration. We tried to focus all discussions, critiques and new ideas on the prototype itself instead of having abstract technical discussions that did not always contribute to the process. We modified or tweaked the prototype as our ideas became more clear and precise. By placing the prototype at the center of the interdisciplinary process, by making it the focus of all efforts, it was easier for both scientists and artists to know where things stood at all times. If we couldn't implement a specific idea in the prototype or if it didn't work in the prototype it basically meant that we had to keep trying. During the development stage of the *Earth Girl Volcano* game we would demonstrate the latest prototype of the game to groups of scientists by playing it and projecting it on a large screen. This way everyone could see it and analyze it in detail. Before and after the public demo to the scientific collaborators we would make game prototypes available on digital tablets for those interested in spending more time examining the latest features. The review by the scientific staff was important to make sure that all the critical preparedness and response features were included in the game and/or were implemented correctly.

**1.6 Quality Control and User Testing**

In addition to the formal reviews of the game prototype by the scientific staff we also conducted game playability testing. This quality control effort routinely tested new and existing game features in the prototype, created bug reports for items that needed to be fixed, and made playability suggestions. The core internal testing group was composed of a dozen student interns from a variety of fields of study including earth and environmental science, animation and gaming from Nanyang Technological University and Nanyang Polytechnic. We also relied on outside testers who volunteered their time to play the game and provide feedback in the areas of playability and comprehension. Many of these volunteer testers were preteen and teenage *digital natives* who are familiar with a variety of interactive game genres: their feedback was invaluable in fine-tuning the mechanics of the game, and in identifying the strongest features of the game and those that required improvement. We worked with volunteer testers in Singapore, Indonesia, Philippines, Vanuatu, China, Italy, Mexico and the United States. We also received useful playability suggestions during our PICO oral/poster interactive demonstrations at the 2017 and 2019 EGU conferences in Vienna, Austria.

**1.7 Interdisciplinary Collaboration**

Inter-disciplinary collaboration between scientist and artists is a fundamental part of developing a game of disaster prevention, and this presents unique possibilities and challenges. The idea of inter-disciplinary collaboration has wide appeal but making it work in reality can oftentimes prove more challenging than expected. Scientists and artists are used to different methodologies and usually operate under different parameters. The scientists who participated in this project offered a high

degree of expertise but oftentimes were uncomfortable with a game development process that seemed to them too intuitive. Artists were at times uncomfortable with a scientific method that seemed rigid and points of view that seemed narrow-minded. In order for the collaboration to bloom the team had to develop a mechanism for collaborators to maintain an on-going dialogue and focus on the end result. The team had to find and practice a methodology of interdisciplinary collaboration that was suitable for this project. Exchanging points of view was an important part of the inter-disciplinary collaboration, and we believe this broadened the horizons of most participating scientists and artists. The former were exposed, for example, to the challenging and complex process of developing and producing an interactive game with high-production values. The later were exposed to the complexities and technicalities of natural hazards. Oftentimes some in the scientific community assume that making a game, or "gamifying" a particular subject as it is sometimes called, is a straightforward process mostly focused on scientific information and software technical issues. Likewise many in the gaming community don't always value the critical importance of deeply understanding the scientific and technical data as a requirement to produce gaming experiences that provide the correct essence of the subject at hand. Our empirical research and practice in making games about disaster preparedness has taught us that an open dialog and discussion between scientists and artists is critical when making games about disaster preparedness and response. Only a true and deep collaboration between game artists and Earth scientists is able to produce games that non-technical users play because they want to not because they have to, games where true learning is embedded in gameplay.

## 2 Act One, Connecting with the Community

Each dramatic act in *Earth Girl Volcano* functions as a unique moment in the overall process of preparedness and response, and in this section we analyze each of the three acts in the game. Most traditional theatre plays or films are divided in several acts, usually three (McKee, 1997). This is meant to separate different blocks of the story, and introduce large shifts and twists. Interactive games are not plays or movies but they also tell a story, and many of the same techniques and conventions used in theatre and film can be considered when designing a game. We loosely structured the interactive game following the traditional three-act structure, with each act providing a unique dramatic purpose. Act I in *Earth Girl Volcano* sets the stage for action, Act II provides an opportunity to explore and understand the challenges faced by the community, and Act III provides an opportunity to actively try to save the community from disaster. Writing for games is a bit more technical than writing for films, for example, because many hardware and software technical issues must be considered and built into the script itself, something that is rarely done in film. The somewhat technical script used to write games is known as a game document and, just like in film or theatre, the actions specified in the game document need to be staged. Any seasoned theatre or film director knows that there is a long road between a good screenplay or script and a good play or movie. The same is true for games and game documents. A script after all is words on a page, a collection of unrealized characters, dialog, locations and action. A script needs to be staged before it can be turned into an experience for the audience.

Generally speaking staging refers to the arrangement of actors, props, lights, cameras, timing and action on the set. The above is also true for games, the art of staging in game design plays a critical role in how the story is told and how the players are allowed or encouraged to participate. Every moment in an interactive game needs to be considered and staged with a purpose, a purpose that is meant to create an emotion and/or to convey an idea. In the case of *Earth Girl Volcano* we intend to convey basic concepts of volcanic hazards and disaster risk management. Before an interactive game can exist its game document, as game scripts are often called, needs to be skilfully translated into scenes, performances, moments of action and reflection, and opportunities to interact. Players of an interactive game want to feel emotions when they play a game and, generally speaking, good staging is directly proportional to the emotional impact of a game.

The basic setting in Earth Girl Volcano, and the gameplay itself, is that volcanic activity is increasing nearby and the player has the opportunity to save as many people as possible by using a variety of resources. The premise becomes evident on the welcome screen where the player is encouraged to gather facts from the local inhabitants (Fig. 3).

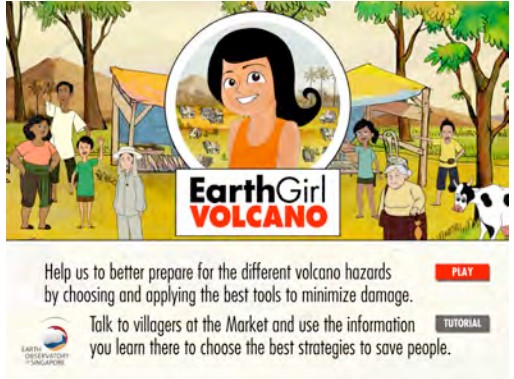

Figure 3: The welcome page encourages players to interact with members of the local community.

The game has five major areas that are seamlessly connected to one another, and each area has a dramatic purpose. These areas are the Map, the Market, the Toolbox, the chosen location (in either Exploration or Action Play modes), and the post-game Feedback and Statistics (Fig. 4). In terms of dramatic staging we can think of these game areas as acts in a play or a movie, and each act has a dramatic purpose and a possible outcome. Through his or her choices the payer can emerge as hero or as villain. An optional tutorial is also available at the start of the game.

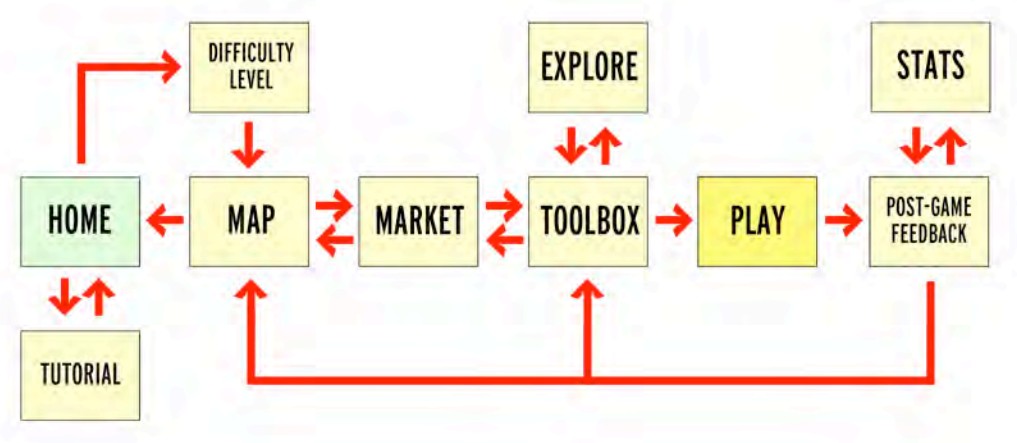

Figure 4: Flowchart showing the primary flow between the areas of this casual game.

Act I of *Earth Girl Volcano* starts with the welcome page but quickly moves to the Navigation Map. In Act I the protagonist, the player, must make two important choices based on his or her confidence to defeat the impending disaster. These choices include one of three possible difficulty levels of gameplay and one of nine possible locations where the action will take place. This early and somewhat blind choice also highlights the fact that the player is getting into some sort of an adventure where not all the parameters are known before hand. It can also be inferred from this suspenseful moment that some effort will be necessary in order to find out what is necessary to succeed in the chosen location. The level of difficulty chosen greatly influences the intensity of the challenges faced later in the game.

The fact that players must make an important choice in Act I but have not had a chance to learn much if anything about volcanic hazards is by design. The goal is to introduce the concept of uncertainty, which is an integral aspect of disaster preparedness and response (Stein, 2012; Tsai, 2015). Uncertainty is present even when robust systems of volcanic monitoring are in place, as it is impossible to predict the exact moment of a volcanic eruption based on a sequence of precursors and statistics. The level of uncertainty in locations lacking monitoring and early warning systems becomes even more serious, and the community's speedy reaction becomes particularly critical. Sometimes in real-life situations of disaster risk management (DRM) we must make educated guesses and take calculated risks, with the understanding that the more we know about the challenge at hand the more prepared we will be and the better our guessing is likely to be. Strictly speaking the staging of the action in the *Earth Girl Volcano* game starts in the Map, where the player is placed in the position of the protagonist. This is a suspenseful start with many unknowns for the player to really know what to expect (Fig. 5).

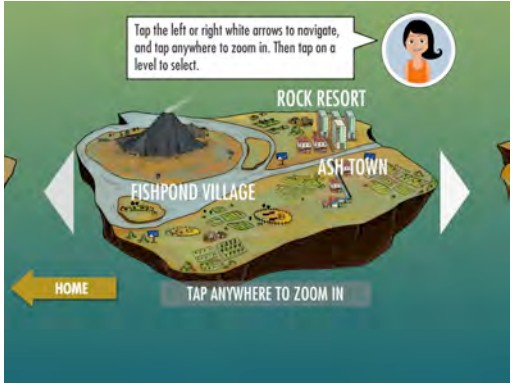

**Figure 5: The Navigation Map displays nine different locations
where the action can take place.**

### 2.1 Levels of Gameplay Difficulty

*Earth Girl Volcano* offers three levels of difficulty: beginner, intermediate and advanced. The pedagogical goal behind
offering three significantly different levels of gameplay difficulty at the start of each round of play is to allow the player to
quickly make an assessment of his or her own skills and ability to stomach more or less risk. The game allows players to
choose any level of difficulty in any order without limitation. It is possible to play a location in Advanced mode first and in
Beginner mode second. This freedom of choice intends to give players a feeling of control on what they play and how they
play it. This characteristic also emphasizes quietly that the game is about exploring and learning to defeat volcanic disaster
rather than accomplishing a high numerical score. The post-game section provides qualitative feedback and detailed
performance statistics (i.e. people saved, casualties, tool usage, etc.) but it does not focus intrinsically on a quantitative final
score. The goal is to be prepared for every possible hazard no matter what the level of challenge, and every hazard counts the
same when it comes to being prepared and aiming for zero casualties.

The Beginner level provides simple challenges, generous budgets and ample time, providing single-hazard scenarios and
gameplay that are well suited to non-experts or first-timers. The Intermediate level offers an increased amount of multiple
hazards per location, this provides exciting yet challenging gameplay and it requires a judicious choice of tools. Budgets and
timelines are somewhat tight and players are forced to carefully strategize in order to succeed. The Advanced level is the
closest approximation to a real life situation where it seems that there are never enough resources to cleanly defeat a multi-
hazard challenges: not enough funds to be perfectly prepared not to launch a perfect response, higher population densities,
and multiple hazards with a high degree of intensity. Much like real tourist locations in the Chilean Andes, for example, the
Rock Resort location features a large tourist population that is largely unaware of the nearby volcanic risks. The advanced
gameplay in *Earth Girl Volcano* has the feeling of a real emergency more than a game: saving everyone cannot be taken for
granted.

### 3 Act Two, Preparing for Battle

After choosing a destination for the adventure the player is gently guided to the Market for a reason: because getting to know
the local inhabitants leads to increased knowledge and understanding, which in turn increase the player's ability to save the
community (Haynes, 2008). Act II is split between the Market, the Toolbox and the exploration of the location itself. Act II
could be summarized as "preparing for battle."

### 3.1 The Market and the Characters

The Market provides a lot of the backstory, and players with limited information about volcanic hazards can only benefit
from paying attention to this part of the game. This area is meant to foster learning by oral storytelling by having player
"listen" to the villagers' experiences and opinions. The Market also provides clues regarding the ideal tools for each location

and level of gameplay difficulty. During testing with our youngest audiences, 6-8 years old, we methodically observed that they tended to read every single dialog available because they understand intuitively that this will raise their chance of survival. Many of our teenage test subjects stayed in the Market only briefly or skipped it altogether, they were anxious to get to the action and try their skills even if that meant that they would have to try several times before reaching success. It is in the Market where the protagonist can invest energy to find out about the challenges faced by the community and is able to start putting together a strategy. Some of the villagers in the Market have useful information about how past occurrences of the hazard have unfolded, plus strategies that have failed or succeeded. Others may have strong opinions about what to do or not to do in case of volcanic activity or a possible evacuation. So it is in the Market where the player can increasingly assume his or her position as potential hero. The player, now clearly the protagonist, can decide whether to talk to the villagers or not and, like in any good drama, this choice will have consequences (Fig. 6).

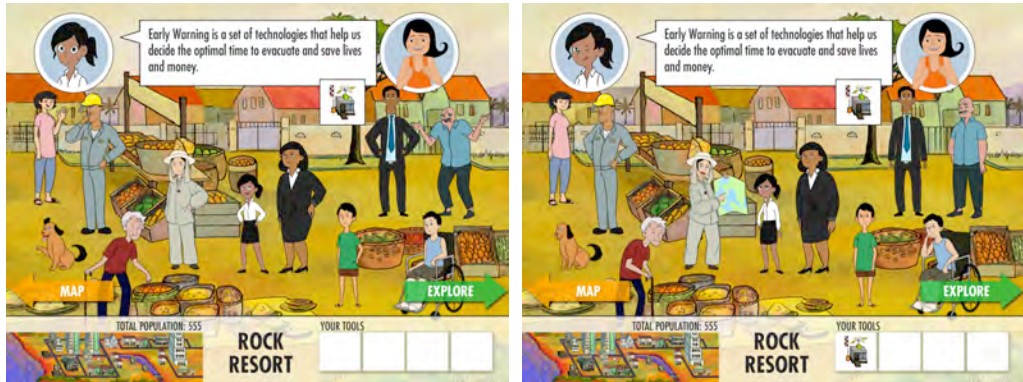

**Figure 6: In some of the Market interactions the villager's statement suggests using a specific tool, and her facial expression changes when the player agrees to add the tool to the temporary toolbox in the lower right.**

The cast of characters in the game includes over a dozen individuals, each one with a unique personality, set of circumstances and opinions. Developing and including unique characters in the game is important because it reminds the player of real people and this facilitates for players of the game to identify with some of the characters and the situations they face. It is easier for players to create an emotional bond with specific personalities and opinions than with an anonymous crowd or a collection of inexpressive characters (Vikan, 2017). Few disaster games if any, to our knowledge, make consistent use of characters with distinct personalities and opinions, deep beliefs, traditions and even misconceptions.

The stories told by the characters in the game are an integral part of the storytelling about volcanic hazards, and they connect the player with real experiences (Gottschall, 2013). Some characters in the Market provide useful factual or historical information while others may provide opinions that may be incorrect or misleading. The dramatic purpose of this incorrect information is that the player must decide what information to use and what to discard. The mechanism of providing incorrect information places the player in a highly proactive position of having to constantly evaluate information, prioritize it and turn it into actions. From the dramatic point of view featuring a character that provides misleading information only makes the game more realistic and hence more effective. The point of the Market dialogs is not to lecture the player about what is scientifically correct but to engage the player's mind into considering different statements about volcanic risk and hazards, evaluating them and making a value judgement based on the player's knowledge at that moment. The incorrect information occasionally provided by some villagers also puts in evidence that the ability of the player to make correct decisions is directly proportional to the player's knowledge, this can quickly act as a motivator to spend some time talking especially to members of the local community who seem to be able to provide useful tips. The game cannot control whether a player decides to investigate and explore but it can provide a situation that is conducive for that to happen. We wrote hundreds of Market dialogs based on real experiences, dialogs that reference the volcanic hazards, the tools provided to address them, and the personalities of the characters in the game. The small sampling of ash fall related dialogs in Appendix A gives a sense of the general tone and style.

The ways the characters look and move are meant to express their personalities and attitudes (Thomas, 1995). The cast is meant to present a variety or personalities, ages and ethnicities that can easily work in multiple locations. We did extensive research to develop character designs that were appropriate and believable within the context of a disaster casual game. With so many possibilities and limited resources we decided to start with a basic group that would look natural throughout Southeast Asia, then we fine-tuned it so it would make sense in a variety of locations along the Ring of Fire. This basic initial cast includes a baby and grandparents, young people and adults. The initial cast also has rural, factory and city types, and the skin tone ranges from light to brown. During a later stage we created an African version of the entire group, including matching props for the locations, and found inspiration in the looks found in the regions surrounding major volcanoes in that continent. We have started to create a Vanuatu version of the cast and props (Fig. 7).

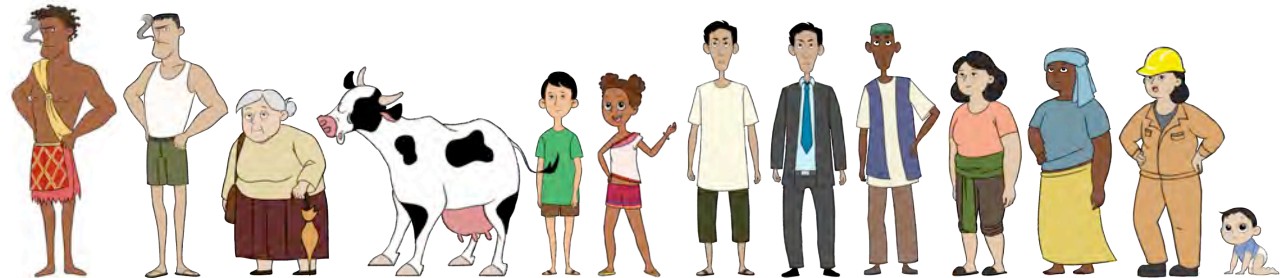

**Figure 7: Line-up of sample characters for different versions of the game (L to R): Street Guy (Vanuatu and original), Grandmother, Cow, Good Boy, Teenage Girl (Vanuatu), Father (rural, city, Africa), Auntie (rural, Africa, factory) and baby.**

In addition to the character's dialog in the Market we also use their facial expressions to communicate their personalities and attitudes to the player. The same approach is used in the Post-Game Feedback area at the end of the game. Emotional connection is a key objective and a key driver of *Earth Girl Volcano*, and one of the implementations of this idea in the game is in the form of the character's facial expressions displayed in the Market and in the Post-Game Feedback sections. These facial expressions are used to provide feedback on the strategic choices made by players and are matched to different character statements. The facial expressions in the game echo the character's written dialog, often times in a humorous way. It is widely accepted that facial expressions are effective conduits for emotional feedback (Ekman, 2007), and we developed for each character sets of the seven emotions that are widely used throughout the character animation industry. These seven emotions are happiness, sadness, surprise, anger, fear, contempt, and disgust. In addition to the basic emotions we also developed tonalities of each emotion to further the emotional precision of each character statement or response (Fig. 8). These tonalities include degrees of an emotion, for example, as in neutral happy, happy, very happy, nice!, good job!, and that's great! happy.

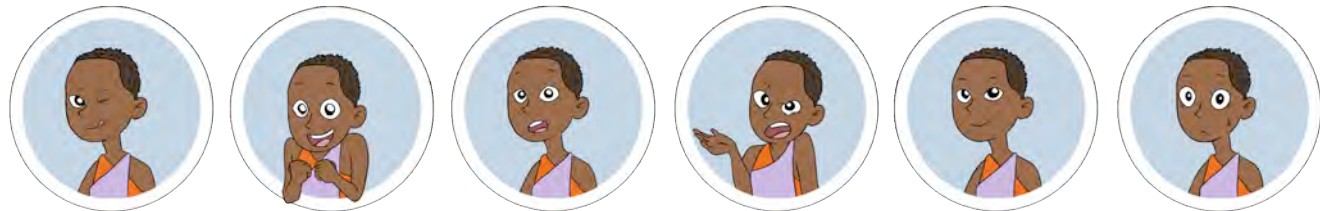

**Figure 8: Facial expressions used for the Teenage Girl character in the African edition of the game, each representing a specific emotional tone: mild happiness, excited happiness, mild surprise, disgust, quiet happiness, and fear (L to R).**

**3.2 The Toolbox**

The Toolbox provides sixteen tools that can be used to prepare for, mitigate or respond to volcanic hazard. The tools are organized in three different categories: infrastructure, technology, and education and services. Players can choose a maximum of eight tools and their choices are shown as Active Tools. When selected every tool displays critical information such as cost and a simple description of its functionality. The Early Warning tool can only be used once during gameplay but

others can be used multiple times as long as there are sufficient funds. The Toolbox is like the brain and the hands of the game because most of the possible mitigation and response strategies in the game are here (Fig. 9). This is where the player makes his or her first round of major choices, and the chosen tools eventually determine the outcome of the emergency. After playing the game a few time players realize that success in *Earth Girl Volcano* is all about the chosen tools, their 410 sequencing and the speed with which they are applied to the emergency.

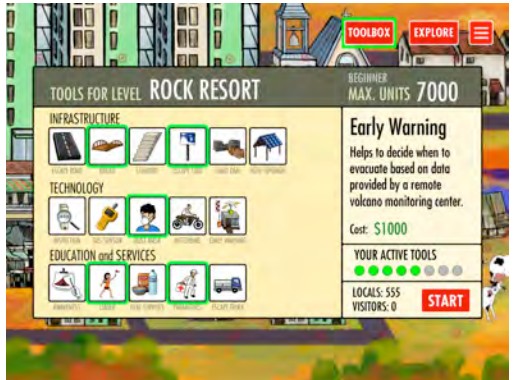

**Figure 9: The Toolbox includes sixteen tools that can be used to address**
**issues of mitigation and response to volcanic hazards.**

The principles that guided our development and selection of the final tools and their mechanics are based on practical issues 415 of mitigating and responding to volcanic hazards. Some of these principles about tradition, perception, practical issues or culture, for example, are explored here. Communities at risk oftentimes have preferred solutions when it comes to hazard mitigation and response. Tools that might work in one culture might be inappropriate in another. Sabo dams for example, also known as check dams, are widely used in some countries but not favored in others because of their complex design and engineering requirements (Nakatani, 2008). Likewise, swimming across rivers during evacuations is also not recommended 420 in locations where the geography might create dangerous river currents (CDC, 2018; Ready, 2020). Inexpensive tools can sometimes be highly effective. Both the Awareness tool and the Community Center address this issue: knowledge is shared with the entire community in an informal and low-cost setting (Red Cross, 2020). This is also the case, for example, of easy to fabricate evacuation signs that point crowds in the direction of shortest or safest routes. Without these signs individuals in a panic or unfamiliar with the environment are unlikely to find their way in time. The Awareness tool in the game is another 425 example of another low-cost and possibly the best-ever preparedness tool: education. In certain situations expensive solutions are not only useful but are necessary to preserve life. Early warning systems, for example, are indispensible in scenarios where a few minutes can make the difference between life and death. Volcanic hazards impact communities with extensive or limited resources and therefore overall solutions must include a range of tools from low cost to expensive. That is why the game's toolbox has solutions within a cost range. Volcanic events often create multiple hazards, some small some 430 large, and the best way to handle this is by attacking each hazard with the proper tool. There is no single tool that can take care of multiple-volcanic hazards in one swoop: small hazards can be mitigated separately from large ones. Dust masks, for example, can minimize the effects of ash fall on the population but a full evacuation is necessary to survive a burning cloud or pyroclastic flow. Last but not least amplifying the existing community-based proven solutions is sometimes the best solution. That explains why it is important "talking" to the characters in the game's Market.

Initially, we came up with over thirty possible tools to address different aspects of volcanic risk. But after much trial and error and significant user-feedback early in the process we decided to keep the toolbox as compact as possible. Developing the tools, fine-tuning their mechanics and structuring the toolbox were one of the most interesting aspects of this project but we shall limit ourselves here to describing the sixteen tools included in the final toolbox. We did early testing of the prototype with about twenty individuals. More than half of those users during early playability and comprehension testing 440 seemed overwhelmed by toolboxes with more than a dozen tools, and their gameplay would become slow and inefficient. Others seemed to lose interest in using tools that were too technical or that required a high degree of expertise to be used

(Ibrahim, 2014). Based on the test users' written feedback we decided that in the context of a casual game targeted at a mainstream audience it didn't make sense to have every single possible tool. This decision was amply validated while testing prototypes that incorporated user feedback. The final set of tools includes six Infrastructure tools: escape road, bridge, stairway, escape sign, sabo dam and roof upgrade. The Technology tools include: inspection, gas sensor, dust mask, motorbike, and early warning. Last but not least the Education and Services tools are: awareness, leader, evacuation supplies, paramedics, and escape truck. The plain language descriptions of the final tools used in the *Earth Girl Volcano* are in Appendix B.

Generally speaking, a specific tool in *Earth Girl Volcano* is meant to address a specific action or a specific hazard, for example sabo dams are only used to mitigate volcanic mudflows. But volcanic disasters are multi-hazard events and in many instances a hazard may require multiple tools to address its different aspects. In the case of ash fall, for example, three tools can be used to respond to it directly: the dust mask, the roof upgrade, and inspection. The dust mask is a low-cost and highly effective tool in the game that can be applied to multiple individuals at the same time, and instantly protects their health. During episodes of intense ash fall the health of villagers in the game decreases rapidly, sometimes leading to collapse and possibly death. Mandated evacuations in game locations that are dominated by ash fall are likely to result in some human and cattle casualties unless dust masks are used. The roof upgrade, another tool related to ash fall, fixes damaged roofs to avoid collapse from heavy ash deposits. Some of the damaged roofs, especially those of huts or small houses, can be easily spotted but the Inspection tool is necessary to make sure that a structure is damaged and requires an upgrade. The pedagogical purpose of the Inspection tool is to highlight the fact that specialist knowledge and significant expense are sometimes necessary to retrofit communities against disasters (Jenkins, 2014). In some instances a single tool may be used to directly address multiple hazards. The evacuation sign, for example, is an inexpensive tool used in the game to address ash fall and other volcanic hazards. This tool attracts individuals with a high level of volcanic hazard awareness who are in the process of evacuating to fast and safe evacuation roads, and speeds up their escape from high danger. In general the available game tools can prevent or mitigate the physical impacts of most volcanic hazards. Burning clouds (pyroclastic flows) are the exception to this rule because their physical impacts cannot be mitigated for the only alternative is to issue an alert and carry out an early evacuation.

### 3.3 Exploring the Locations

*Earth Girl Volcano* offers nine distinct locations where the action takes place. The aim is to design sets where volcanic hazards can unfold in a wide variety of possible ways and permutations. Each of the locations is designed to facilitate specific situations that are part of the gameplay, and they are named in obvious ways: Pyro Creek, Pyro City, Mud Village, Rock Resort, Ash Town, Fishpond Village, Lava Buildings, Petrol Hell, and Gas Village. These obvious names provide first-time players with a small clue about a possible critical gameplay in that location. The nine locations provide rural, urban and industrial settings, some have basic infrastructure while others are in remote locations (Figs. 10 and 11).

None of the locations in themselves are more or less challenging than the others. It is the combination of multiple variables that yield a high-risk environment, including topological features, props, population and level of gameplay difficulty. Each of the nine locations has built-in features and also passive props and inhabitants that can impact the result of a potential evacuation. The built-in features are literally painted on the background, and the level designers place the passive props in strategic locations. Some of the built-in features include, for example, proximity to the volcano, evacuation aids (i.e. paved roads), evacuation obstacles (i.e. rivers), and people. Some of the passive props that are part of the mitigation and evacuation dynamics include huts, houses and buildings. Some of the tools in the toolbox are also props that are activated when the player places them in the environment. Some of these active props include sabo dams, escape trucks, motorcycles, and the early warning system.

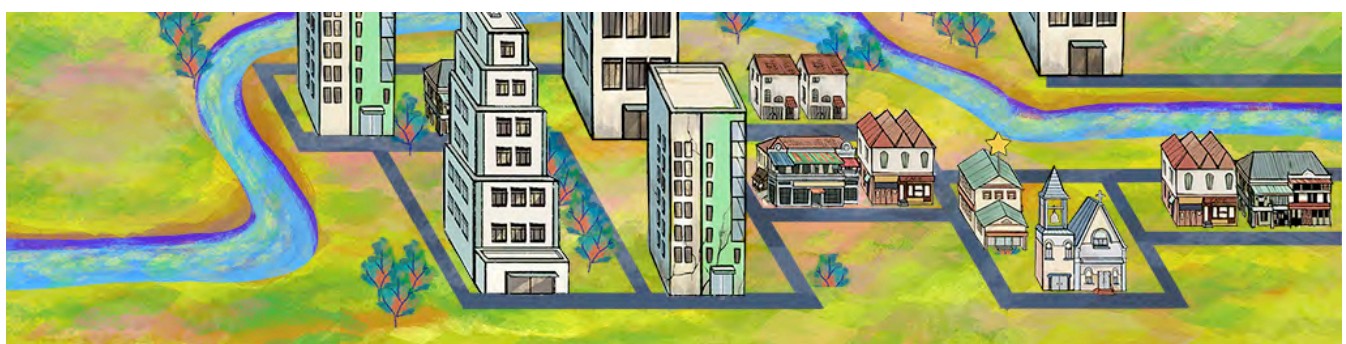

**Figure 10: Example of an urban location in the game, Pyro City,
with painted background and overlaid props.**

Each of the nine locations has obvious topological features that can be easily identified while in the Exploration Mode. These topological features along with the human-made structures, or props, hint at unique opportunities or obstacles in case of an emergency evacuation. Each location includes features that are inspired by real places and situations faced by real communities living next to or near a volcano (Oppenheimer, 2018). The Exploration mode provides the player with an opportunity to assess the topological features of a location, and quickly incorporate that knowledge into the crafting of a possible mitigation and response strategy. The nine locations in the game are described in Appendix C.

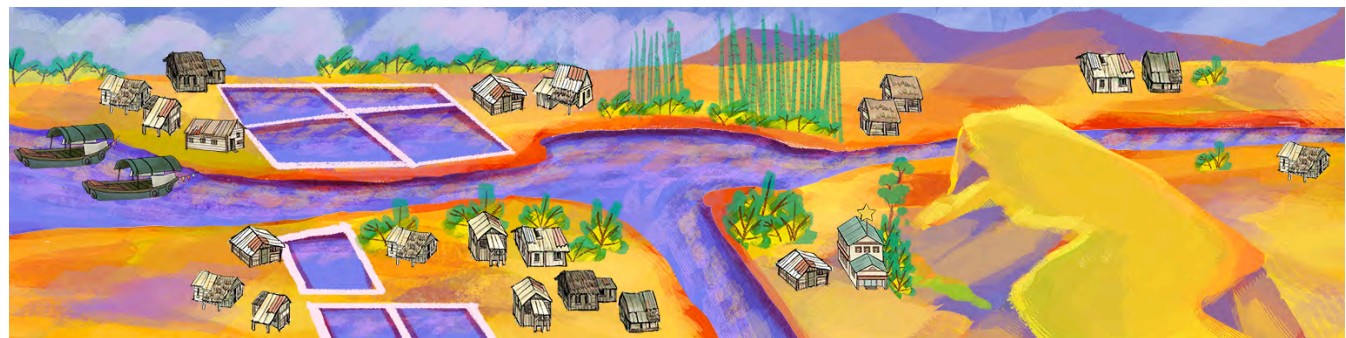

**Figure 11: The Gas Village is an example of a rural locations in the interactive game.**

In addition to the obvious obstacles and opportunities there are other features and clues that are not as easy to spot as a hill or a river. These hidden factors, or hidden variables, also play important roles in the overall development of the disaster and they include primarily the people. "Where are the people? Where is everybody?" users kept asking while exploring locations during early game testing. It is striking, especially for first-time users, that the locations seem a bit empty of people before the action starts. While the toolbox may display a number of inhabitants in the hundreds, the streets and the fields initially look a bit empty. This is because many are actually indoors, we don't see them but they are there: kids at school, and adults at work or at home. This staging of the action is a hint to emphasize that one of the goals of the evacuation is to get people out, to convince them to leave. It is well documented that evacuees do not always leave their homes behind happily or automatically (Vaessen, 2017). In addition to the local population there also are visitors, outsiders who are not on-site at the start of the volcanic episode but who are driven away from their towns and seek safety by the game location. This staging is to emphasize that it is not always possible to have a perfect plan, and that major unexpected events might and will impact the development of what seemed to be a "normal" evacuation. The message to the player is that normal evacuations do not exist in disaster mitigation and response, every disaster cases poses unique challenges and those in charge must be on constant alert.

**3.4 Custom Character Behaviors**

A simulation engine runs the *Earth Girl Volcano* game, and this type of software allows for every villager to have unique characteristics that can be defined by a few variables. Some of these are assigned by default, others are stochastic and a few are context sensitive. The top four variables that define how a villager in the *Earth Girl Volcano* game reacts during a volcanic emergency are health, awareness, "stupid behavior," and swimming ability. These variables can render behaviors

that are faithful to the myriad ways in which different individuals behave during a volcano disaster. This in turn brings the behaviors and mechanics experienced in the game closer to what real people experience during real volcanic events, it makes them more believable (Rooney, 2012). Health, for example, can be thought of as battery power, and villagers start with a certain amount of health based on their age and physical characteristics. Health in the game decreases naturally throughout time but certain hazards like ash fall can have a devastating effect on it, dust masks prevent or stop this condition. Awareness is a sort of disaster know-how factor that allows villagers to make the best choice in a variety of situations. Volcanic hazard awareness is represented with a color circle around the villagers' heads: red represents a lack of knowledge, orange represents limited knowledge, yellow represents some knowledge, and green represents maximum knowledge. Villagers with maximum awareness, for example, automatically start to evacuate no matter what when the Alarm Level reaches Yellow. The amount of disaster awareness is distributed throughout the population is a standardized way and most characters have a default awareness ranging from low to medium. The Awareness tool can raise practically everyone's awareness level to the maximum and the results are highly noticeable. Any changes in disaster awareness have a decisive impact on crowd behavior and the way in which an episode of volcanic activity plays out. Changes in the health and awareness values are time-sensitive: raising it at the start of gameplay has a different result than raising it in the middle of gameplay. We use the term "stupid behavior" in the *Earth Girl Volcano* game for actions that go against all common sense, and only the Leader tool can remedy this situation. Examples of this behavior include running towards the flowing lava or refusing to evacuate when everyone else is evacuating during a red level alert. The last of the hidden variables in question is swimming ability and it allows villagers to cross small bodies of water even when bridges are not available. In some real situations, such as the Chilean Andes, this feature strictly speaking is not as relevant due to geography. Values for both stupid behavior and swimming ability are statistically assigned to the population by default.

## 4 Act Three, Action Under Pressure

Act III in *Earth Girl Volcano* is all about saving the community from a likely volcanic disaster. The player is placed in the role of the protagonist and is expected at this point to have a strategy for mitigation and response if necessary. But players only have limited information and they must act quickly and effectively on the basis of the known facts. They must also remain vigilant and react to any new developments. Some data is displayed in the style of a Heads-Up Display (HUD) and this can help the player to keep track of progress. But regardless of the known information there is always some degree of uncertainty and unexpected surprises. The basic information displayed includes Time Left, Budget Left, and People Saved, and it provides the player with both a sense of urgency and accomplishment. If the player chooses and activates the Early Warning tool the HUD will also display a color-coded Alert Level, an indication of the areas at highest risk (a simple hazard map), and the total cost of a mandated evacuation (Fig. 12).

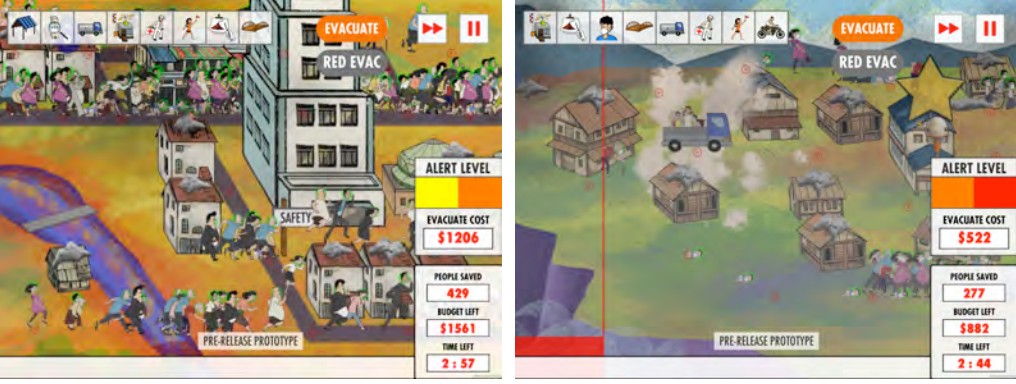

**Figure 12: Quantitative data is displayed during action gameplay to help players assess their progress. Using the Early Warning displays the alert levels and the simplified hazard map.**

A real-life episode of volcanic activity that ends in a volcanic disaster can usually take hours, days or weeks. But because *Earth Girl Volcano* is a casual game we compress the duration of each action cycle of volcanic activity to between three and

five minutes. Everything unfolds very rapidly and demands full attention and swift action. One of the results of staging the action in such a compressed way is the feeling of stress by the player, too many things happen at the same time and too many things need to be taken care of. The pressure, the uncertainty and the overload experienced by the player at this stage bring the experience close to what an emergency evacuation manager might feel in a real emergency situation. Once again in the game the staging of the action is designed to provoke in the player an emotional connection with the moment, with the act of trying to save a community from a volcanic disaster.

The experience of playing for the first time a single location in the *Earth Girl Volcano* interactive game can take an average of 8-15 minutes depending on how much time a player, or a group of players, spends in the Market and exploring the location before triggering the volcanic activity. We conducted playability testing with volunteer students or interns at the Nanyang Technological University in Singapore, and one of our key on-site testing locations was at Malapoa College in the city of Port Vila, in Vanuatu a nation with a high-density of active volcanoes (Fig. 13).

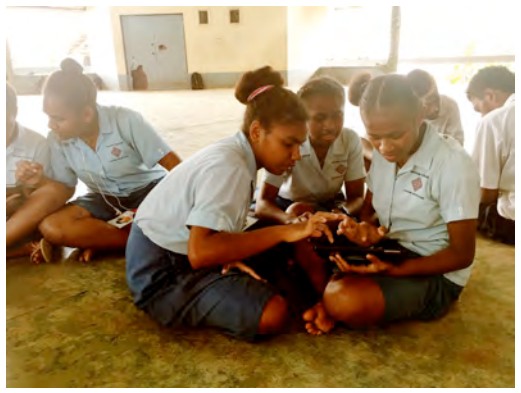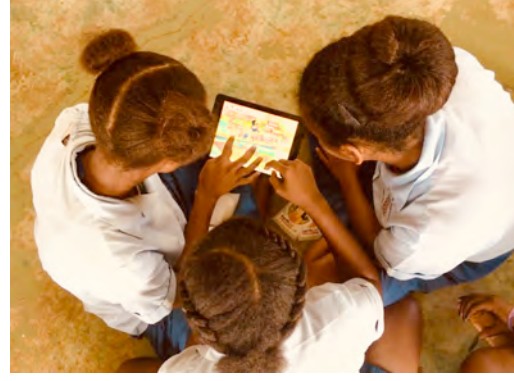

**Figure 13: Malapoa College students in Vanuatu test play *Earth Girl Volcano* on a tablet and discuss possible strategies for a specific location. Photo by I. Kerlow.**

The locations in the game can be played in multiple ways, and during testing we recorded these behaviours casually as to not to interfere with the spontaneity of playing the game. We have observed players that start by reading just a couple of dialogs in the Market, go straight to the action and play the same location over and over in Beginner mode until they are able to save practically everyone. We have seen players who try different locations and even if they get a low score move onto other locations before returning to a favorite location and playing that over and over at different degrees of difficulty. Most of the times that we visited schools to demonstrate the game we played in teams of two to four students, each team with a tablet. In this variety of team playing the players usually read every single Market statement, talk about it and try to identify the takeaway idea so they can incorporate it in their mitigation and response strategy. Playing a single level in team mode may take 20 minutes because of the extra discussion but it is extremely satisfying to watch a group of young players argue about their ideal strategies for a zero casualty result. To wrap-up Act III the game offers general qualitative feedback on the player's performance, focusing on his or her strategic top accomplishments and worst failures. Numerical statistics are also available for players to review after playing each cycle of volcanic activity (Fig. 14).

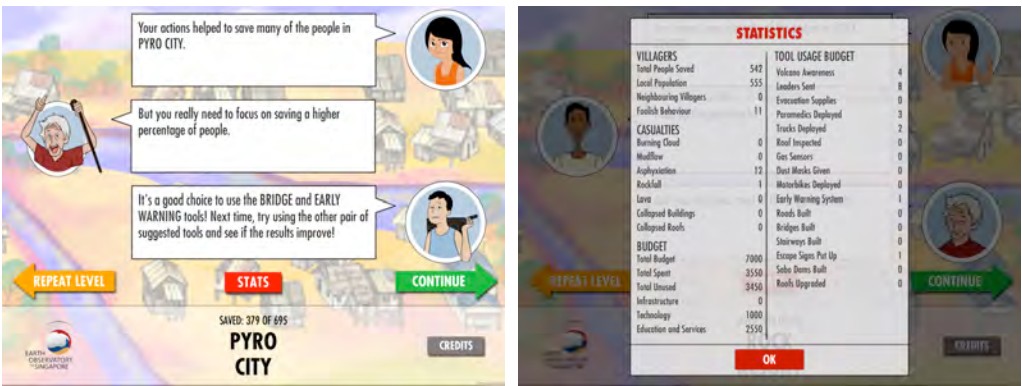

**Figure 14: Players can review general feedback on their performance and strategic choices (left), as well as gameplay statistics (right).**

# 5 Characterizing Volcanic Hazard Complexity

Our ultimate goal in characterizing volcanic hazard complexity for *Earth Girl Volcano* was to have non-technical audiences identify with communities at risk from the impact of volcanic hazards. We believe that this emotional connection makes it easy for players to become familiar with technical and practical knowledge, and we followed a few game design approaches to increase the potential of emotional connections. These specific design techniques facilitate the characterization of volcanic hazard complexity for non-technical audiences in the context of a casual strategy game. We followed and refined these guidelines from the start of the process and throughout the development process. The techniques include using show and tell (described above in 1.1), custom character behaviors (described in 3.4), and general principles; generalizing from critical details, explaining and dramatizing, using humor, and combining simulation and scripting.

## 5.1 General Principles or Golden Rules

In the context of volcanology scientists approach the details of specific episodes of volcanic activity by applying the scientific method: observing the phenomenon, analyzing it using a variety of theories, and proving the results in the laboratory. But using the same approach in the context of communicating volcanic hazards to the general public proves ineffective, as mainstream audiences lack the specialized knowledge and/or interest to fully understand and use such technical information (Bellotti, 2011). In *Earth Girl Volcano* we use general principles that are based on practical experience, also known as golden rules or rules of thumb. Because of their simplicity and their direct link to empirical experience these principles are generally useful for presenting the essence of volcanic hazards and their impact on communities. Some of the broad principles presented in the game are explicit and others can be inferred from the experience of gameplay. Many of the Market dialogs present true information and explicit rules of thumb that encapsulate the experience and prior practice of real communities that are represented by the characters in the game. Many of the volcanic precursors shown during gameplay in the locations show actions and events that can easily be generalized into rules of thumb. A few of the empirically-based rules in the game include, for example: ash fall impacts people and cattle in different ways depending on their health level; ash fall can be mitigated by wearing protective masks; the weight of ash fall can collapse the roofs of frail dwelling structures; volcanic mudflows are likely to occur after episodes of intense ash fall and rain; and using an early warning system can provide communities with additional time to respond to an eventual volcanic episode. General principles are efficient heuristic devices that help non-specialists to get a quick sense of complexity and use that knowledge to make swift decisions and solve challenges. Golden rules are broad but true, and easy to remember.

## 5.2 Focus on Critical Details but Avoid Extreme Detail

When it comes to volcanic hazards, the immediate interests of communities at high-risk are the ways that the hazards might impact them. Details such as the precise chemical composition of magma or the ideal arrangements of seismic monitoring devices, for example, are of interest to specialists but less interesting to main audiences because these details are not immediately useful to most of them. Generalizing high-detail information into more abstract information with less detail helps to focus the player's attention on the areas that are of immediate consequence to the well-being and survival of communities. Our own rule of thumb in generalizing volcanic hazard activity in the context of the game is to focus on what is relevant to the gameplay and the game mechanics supported in the game (Fullerton, 2019). In terms or disaster risk management the ultimate goal of a successful generalization is to show the unique dynamics of each volcanic hazard, what to do when an event is triggered, and what are the possible impacts.

Let's take the ash fall volcanic hazard, for example. Players of *Earth Girl Volcano* are shown three important aspects of ash fall: its impacts on human and animal health, on weak structures and roofs, and as a precursor to mudflows. Ash fall presents additional negative impacts in real situations, impacts such as destroying crops or complicating air travel, but we do not

explore those impacts in the game. The goal of the game is not to provide encyclopaedic reference on volcanic hazards but to focus exclusively on the critical details that can be experienced through gameplay.

The variables used to simulate hazards with scientific simulation software can be quite extensive (Francis, 2004) but we had to scale the simulation to the context of a casual game. As a result the volcanic hazards in the game are characterized by just a few critical details that are capable of expressing their essential behaviors. Ash fall, for example, is defined in the game by a few variables including Ash Start Time, Ash Health Damage, and Ash Density. There can be multiple start times for multiple sequences of ash fall, and these are stipulated in the hazard scripts that control gameplay for each location. The

damage variable is used to impact the health of people and cattle over time while they remain unprotected in the ash fall. The numerical value of the Health variable is never displayed in the game and players must keep track of the population during evacuation for any sign of severe health issues that might require a paramedic. Ash density is used to modulate the amount of damage of ash deposits on weak or damaged roofs. To further characterize ash fall we usually specify variables within ranges that are easily recognizable. Ash fall events in the game usually appear as light, medium, or heavy to help the player

quickly assess the most suitable response.

## 5.3 Explain and Dramatize

The act of explaining can provide objective information that helps to understand a specific situation but by itself it does not create emotional connection. Explaining is an important mechanism to communicate scientific topics in general because the point of science is to understand and to gain objective knowledge. But explaining how volcanic hazards works is not

effective in creating the emotional connections that are one of the essential goals of *Earth Girl Volcano*. The cold facts of Geoscience rarely capture the imagination of communities at risk. We have observed and experienced this ourselves oftentimes when talking about volcanic hazard preparedness with individuals from such communities. Dramatizing, on the other hand, is a technique used to create emotion and emotional connections by using characters and the challenges they face in a specific situation. We use the word dramatizing in this instance not as in exaggerating something but as in turning the

human experience into a story, as in staging a situation where characters face internal and external pressures (McKee, 1997). The artists on the *Earth Girl Volcano* team were familiar with a rule of thumb well-known to writers of novels, movies and theatre plays: "dramatize, don't explain." Dramatizing a situation instead of explaining it is a time-proven approach to engage the emotions of the audience. The scientists on the team, on the other hand, were familiar with formulating hypotheses and testing them. The game benefited in the end from each group focusing on what they do best and also trying

to understand how to enhance to what the other group does.

The inclusion of a cute Baby character, for example, is all about dramatizing. Babies are precious beings and most players are concerned with saving the babies in the game. During user testing we methodically observed how practically every player, particularly the youngest, perfected the use of many tools in the game just so that they could save the baby. The mere presence of the crawling babies is a dramatic device that brings humor to the mix and creates an emotional connection

between players and the volcano disaster. The game mechanics of stupid behavior and escape by motorcycle further illustrate the technique of dramatizing. In the case of stupid behavior, described earlier in Section 3.3, the reasons behind it are not explained because they are secondary in the context of an emergency evacuation. In terms of story, an impending disaster represents a life and death external pressure on every character in the game. But characters with stupid behavior respond to their own internal pressures, whatever those may be, and place their lives in jeopardy by refusing to join the

evacuation. At that point we have a dramatic conflict between the desires of those characters and the overall mission of evacuating as many people as possible. The situation cannot be ignored because one way or another it will impact the final score. In dramatic terms this conflict requires resolution and brings the stakes higher. The escape by motorcycle also creates interesting dramatic moments because only the closest characters to the vehicle can ride it, and when placing a motorcycle the player can only estimate who might arrive first. This situation creates suspense, tension and hence, an emotional

connection. The Motorcycle tool is useful for saving stragglers who are usually the elderly, characters with diminished

health and… babies. Volcanic hazards are no laughing matter but fun game design contributes to engaging gameplay (Koster, 2005). Yes, babies in the *Earth Girl Volcano* game can ride motorcycles!

## 5.4 Combine Simulation and Scripting

This interactive computer game makes use of both numerical simulations and script-driven events, both with some degree of stochastic values. This approach is actually borrowed from the creation of special visual effects of natural phenomena where physically based simulations are used to create the overall look and behavior of the material, but a script is used to distribute discrete elements at specific times and locations. This would be the case, for example, of different types of waves that need to be choreographed in a computer-generated ocean. The secondary or "crowd" waves are generated with physically based simulation software but the exact placement and timing of the "hero" wave is controlled with a deterministic script. This combined approach offers both flexibility and the ability to deliver convincing renditions of volcanic hazards. The hazard script in Appendix D paraphrases a simple sequencing of volcanic events in the game that are easy to beat. In *Earth Girl Volcano* we use interactive menus and direct scripting in the Unity programming environment to specify the parameters of each of the twenty seven basic experiences (Fig. 15).

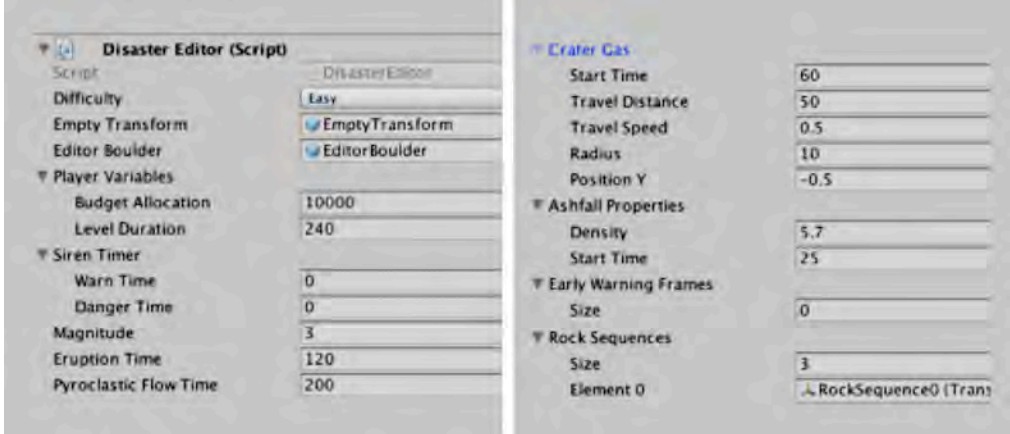

**Figure 15: Detail of a menu used to input volcanic hazard variables in *Earth Girl Volcano*.**

## Conclusions

The process of developing and producing the *Earth Girl Volcano* game in the context of an art/science interdisciplinary collaboration was both challenging and fruitful. This collaboration confirmed that the default methodologies favored by artists and scientist are oftentimes quite different, particularly when developing a science-inspired game. Initially the scientific colleagues were focused on a concept that simulated a volcanic hazard in a most detailed way, while the artistic colleagues focused on a concept that would engage the target audience. Simulation and storytelling emerged as opposing approaches early on, but through extensive inquiry into the goals of the game we managed to develop an experience that combines scripted simulation with engaging storytelling. Interdisciplinary collaboration between artists and scientists when developing a product for a mainstream audience can be a challenging process that requires communication, constant give and take, and where nothing can be taken for granted.

Early in the process it became evident that scientists and artists follow different approaches to research, and that a compromise would be paramount to avoid a failed collaboration. The parameters of modern scientific research have been widely used for over a century. The scientific methodology is narrowly defined and is centered on quantitative data. Practice-based research is another modality of research that also generates knowledge, and it started to be formulated and formalized in the late twentieth century. Practice-based research has become the de-facto research methodology in the creative arts. It can be defined as "an original investigation undertaken in order to gain new knowledge, partly by means of practice and the outcomes of that practice" (Candy, 2018).

One of the motivations behind The *Earth Girl Volcano* was the lack of interactive games that in our initial estimation provided engaging experiences to non-technical users. This original motivation became the goal of our practice-based research, and we developed a methodology to address the challenge. We explored storytelling techniques and crafted a simple way to integrate a story that functioned on top of the underlying complexity and that also provided a simple and intuitive gaming experience. The game incorporates notions of Earth science and evacuation management, but the game itself is neither a scientific project nor an emergency evacuation manual. We believe that engaging science-inspired games must incorporate emotions that players can identify with. These games can help communities to increase their understanding and preparedness and be more than just a routine public engagement exercise. We dissected in this paper our game methodology and the techniques used in *Earth Girl Volcano* to characterize volcanic hazards in an engaging way. We hope this contributes to the active discussion surrounding the nature of practice-based research and the challenges of interdisciplinary collaboration.

The *Earth Girl Volcano* interactive casual strategy game presents a comprehensive range of volcanic hazards in the context of preparedness, mitigation and response. The team was able to create original ways to characterize volcanic hazards within the context of an interactive story that is more engaging than a collection of scientific facts. We hope these characterizations inspire future games and have a lasting impact in the development and production of disaster preparedness and response interactive games. The techniques that we developed and used to convey complexity and to create empathy with the situations depicted in the game have generated multiple inquiries from users and colleagues. The nature of these user requests is a testament to the initial interest sparked by the game, and they focus on possible enhancements and/or modifications to the game. For example: creating game locations that are faithfully modelled after real volcanoes, creating "national" versions of the game where the game locations represent only real volcanoes in a specific country, providing the ability to change and customize Market dialogs, providing the ability for users to edit the volcanic hazard scripts, and adding science-oriented modules with in-depth technical information. These are all intriguing suggestions, feasible within the existing game engine, and are also in-line with the original specifications of the game. Versions of the game customized to a specific real volcano, for example, would pose interesting and unique game design and science communication research challenges. Increased levels of realism or geophysical complexity simulation would likely require a game engine upgrade.

A useful contribution to this project would include in-depth testing to measure the shifts in players' attitudes after playing the game. Another possible next step would be the development and creation of supplementary classroom materials, a task that we initiated but did not complete. Such materials would facilitate the use of the game by teachers in classroom scenarios (Alklind, 2014; Foster, 2015). *Earth Girl Volcano* was translated to Indonesian, Tamil, Spanish, French and Italian. Pending new funding we are keen to continue translating the game to additional languages spoken in the Ring of Fire: Tagalog spoken in the Philippines, Bislama spoken in Vanuatu, Russian and Japanese.

**Appendix A. Dialog Examples**

*Auntie*: Inspect the weak or broken roofs because a small amount of ash fall could make them collapse and injure or kill the people indoors. *Bad Boy*: Breathing volcanic ash for extended periods of time can make us weak and complicate an evacuation. Masks or paramedics do help. *Bad Boy*: We usually get heavy ash when our volcano gets temperamental. We have learned to keep around dust masks just in case. *Bald Man*: Ash fall and smoke are a sign to get ready for a possible evacuation. The signs help people find the evacuation paths faster! *Bald Man*: The rains can easily wash down the volcano slopes some of the ash fall that builds up on the ground during times of volcanic activity. *Father*: The evacuation committee plans to distribute dust masks when the ash fall begins. I don't want my kids to get sick or faint. *Father*: Volcanic ash fall is frequent in this picturesque river town. Some of the roofs are damaged and need to be repaired to avoid collapse. *Fat Uncle*: Heavy ash fall is an indication of high volcanic activity, so be cautious and take action. Better safe than sorry! *Grandfather*: I hope the evacuation committee distributes masks in the event of ash fall. Your nose gets clogged with ash and you can't

breathe. *Scientist*: The amount of ash emitted during a volcanic eruption changes along with volcanic activity. A sudden increase may be a sign to evacuate. *Scientist*: Over time the falling ash is debilitating and may be lethal, especially for the vulnerable. Dust masks can minimize the impact. *Scientist*: Volcanic ash fall is a sign that volcanic activity might increase, possibly with rock bombs or burning clouds moving at 100km per hour. *Scientist*: Volcanic ash isn't the only precursor to an eruption. Toxic gases can also be emitted from the ground before volcanic activity. *Scientist*: Many people live in high-rise

buildings today but we still have to maintain the roofs of short buildings in case of heavy ash fall. *Teenage Girl*: I am on the community evacuation committee, and we have boxes full of dust masks to give away, for use in case of ash fall. *Young Woman*: I worry about the families living in beach huts. Those weak roofs do not withstand the heavy ash we get during eruptions.

**Appendix B. Tool Descriptions**

INFRASTRUCTURE TOOLS: *Escape Road*. The fastest and most efficient way to escape on flat ground. *Bridge*. Escape bridge, useful for crossing wide rivers and lakes. *Stairway*. Fastest and most efficient way to climb steep hills up or down. *Escape Sign*. Signs that direct people to the fastest escape route, they make people run faster. *Sabo Dam*. Barrier that blocks mud flows and slows them down. Multiple sabo dams needed to stop large mudflows.

TECHNOLOGY TOOLS: *Roof Upgrade*, Reinforces damaged roofs to avoid collapse during ash fall. Always requires the

Inspection tool first. *Inspection*. Determines the structural strength of buildings, and must always be used before Roof Upgrade. *Gas Sensor*. Detects gas leaks that may be poisonous or lethal to humans and cattle. *Dust Mask*. Limits damage from volcanic ash but is ineffective against lethal gases. *Motorbike*. Can quickly transport up to three villagers away from danger. *Early Warning*. Helps to decide when to evacuate, based on data provided by a remote volcano monitoring center.

EDUCATION AND SERVICES TOOLS: *Awareness*. Provides four levels of volcanic hazard knowledge, and improves

peoples' evacuation decisions. *Leader*. Individuals who lead people and cattle towards safety. *Evacuation Supplies*. Pre-pays the cost of an evacuation to ensure the best possible one. Cost based on population. *Paramedics*. Rescue and heal villagers who are injured or who become disabled by a volcanic hazard. *Escape Truck*. Provides an efficient way to move up to 20 people and cattle away from danger.

**Appendix C. Location Descriptions**

*Pyro Creek*. This one is a remote location by the volcano. The rustic dwelling areas are split by a river and bound by a lake, and the only escape route is narrow and gets easily congested.

*Mud Village*. A flat rural area located not far from the volcano and split by a few small rivers. The huts and small houses are right by the river.

*Pyro City*. This is a high-density urban area with tall and short buildings. A river runs through the town, and good vehicular

roads are found on the south half of the city.

*Rock Resort*. A tourist resort located by a lake and not far from the volcano. It includes tall condominiums and leisure areas, good vehicular roads, and a channel that partially blocks pedestrian traffic.

*Ash Town*. This small manufacturing city is split by a channel. Small buildings are concentrated near the only vehicular escape route.

*Fishpond Village*. A fishing village by the lake with a few huts and a large mosque, without a vehicular escape route. The environment is fragmented by a system of ponds and irrigation channels.

*Lava Buildings*. A city on the mountains with multiple high-rise buildings, and a single vehicular escape route. Located in proximity to the volcano and split by a river.

*Petrol Hell.* This industrial park and refinery is located next to loading docks, and split by a man-made channel. Good
internal vehicular roads but limited traffic in and out of the fenced compound.

*Gas Village.* A small village in a remote rural area located by large rivers. It lacks paved vehicular roads, and features
multiple obstacles.

**Appendix D. Scripting Hazard Events**

LOCATION: Lava Buildings
DIFFICULTY: Beginner

Budget: 7,500 units

Experience Duration: 240 sec

Population: 425

Incoming Outsiders: 60
Point 1: Start at 130 sec, 20 people

             Point 2: Start at 130 sec, 20 people

             Point 3: Start at 130 sec, 20 people

Evacuation Budget: 3 units per person

Music Change: 120 sec
Early Warning Cut-Offs

             Green: 0 sec. Yellow-Green: 50 sec

             Yellow: 70 sec. Yellow-Orange: 100 sec

             Orange: 120 sec. Orange-Red: 140 sec

             Red: 150 sec
TREMORS: Yes

             Small Tremor-1: Start at 85 sec, Magnitude=0.5, Shake Duration=0.5.

             Small Tremor-2: Start at 120 sec, Magnitude=1, Shake Duration=0.5.

             Large Tremor-1: Start at 150 sec, Magnitude=1

GAS: Yes
Crater Node: Start at 70 sec, People Damage: 0, Distance: 100, Speed: .7, Radius: 10

             Village Node-1: Start at 60 sec, Duration: 15 sec, Radius: 3, People Damage: 0.1

             Village Node-2: Start at 75 sec, Duration 10 sec, Radius: 3, People Damage: 0.1

             Village Node-3: Start at 120 sec, Duration 30 sec, Radius: 4.25, People Damage: 0.1

             Village Node-4: Start at 125 sec, Duration 30 sec, Radius: 3, People Damage: 0.1
LAVA: Yes

             Lava-1: Start at 160 sec, Slow Speed: .8, Radius: .6, Ends at river

             Lava-2: Start at 180 sec, Slow Speed: .8, Radius: .6, Ends at river, Destroy House: 1

ROCKFALL: No. ASH FALL: No. RAIN: No. MUDFLOW: No. PYRO FLOW: No.

**Game Availability**

At time of writing the public can download the English computer version of the Earth Girl Volcano game for Windows and
MacOS from https://earthobservatory.sg/ and https://art-science-media.com/. Several localizations of the game for digital
tablets are also available at Google Play and the App Store. The early Earth Girl Flash games are available at

http://earthgirlgame.com/ and they require the Flash plug-in. We plan to upload the Flash games to the repository website https://itch.io/.

## Video Supplements

Two short videos illustrate the features of this game. The Earth Girl Volcano - Game Overview video (1m 30s) provides an introduction to this casual strategy game. Available at the TIB AV-PORTAL at https://dx.doi.org/10.5446/47697, and also at https://vimeo.com/294752004. The Earth Girl Volcano - Game Demo video (6m 27s) illustrates a sequential gameplay of the game. Available at https://dx.doi.org/10.5446/47698 and also at https://vimeo.com/305156140.

## Author Contributions

Isaac Kerlow conceptualized the Earth Girl disaster games and designed the Earth Girl Volcano game, supervised the development and production teams throughout the process, participated in the creation of artwork, and wrote the original draft of the published work. Gabriela Pedreros and Helena Albert contributed to the methodology during the late stages of production, participated in the validation, and reviewed and edited the published work.

## Competing Interests

The authors declare that they have no conflict of interest.

### Acknowledgements

The core members of the *Earth Girl Volcano* game development and production team made significant artistic and/or technical contributions to the game itself. The complete list of game credits is at www.earthgirl2.com. Design of game and user interface: Isaac Kerlow. Programming: Seah Wen Kai, Zel Ang. Level design: Faye Lim Ying Yuan, Isaac Kerlow, Lisa Lim Yan Xin. Technical artist: Faye Lim Ying Yuan. Lead artist: Nguyen Thi Nam Phuong. Sound and music: Jeremy Goh. Early prototyping: Kim Van. Market dialogs: Isaac Kerlow, Lisa Lim Yan Xin, with additional contributions from Lee Jia Min, Seah Cheng, Yeo Shu Hui. Production coordination: Antoinette Jade, Jingqin Tioranu, Victor Chan. The *Earth Girl Volcano* game was developed in consultation with members of the Volcano Group and other research groups at the Earth Observatory of Singapore, with the active participation of Susanna Jenkins who was the co-investigator of record, Fidel Costa, Caroline Bouvet, Benoit Taisne, Li Weiran Alex, Fabio Manta, Lauriane Chardot, Chris Harpel, Jason Herrin, Dawn Sweeny, Christina Widiwijayanti, Dayana Schonwalder, Gareth Fabbro, Stephen Pansino, and Dorianne Tailpied. Tanslations: Ratih Oktarini, Riko I Made (Indonesian), Elakeyaa Selvaraji (Tamil), Erinna Sacco, Fabio Manta (Italian), Lauriane Chardot (French), Gabriela Pedreros, Isaac Kerlow (Spanish). We thank the following institutions for providing major support: Earth Observatory of Singapore, AXA Research Fund, National Research Foundation, Ministry of Education, The World Bank Group, Global Facility for Disaster Reduction and Recovery (GFDRR), Nanyang Technological University, Nanyang Polytechnic, and Malapoa College. We also thank the many individuals throughout the world who have invested their time, funds and effort to develop and implement this project. Special thanks to Damià Benet, Jacques Frety, Wendy Tan, Simone Balog-Way, Tiziana Lanza, Kerry Sieh, Chris Newhall, Glenda Tapel Newhall, Florian Schwandner, Ben Horton, Adam Switzer, Jegannath Ramanujam, Naveen Raj Kunaseelan, David Higgit, Bruce Malamud, Cheng Hoon Khoo, Esline Garaebiti, Shem Simon, Richard Shing, Henderson Tagaro, Johnson Toa, and Jackie Potgieter.

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
