# Peer review of "Earth Girl Volcano*: Characterizing and Conveying Volcanic Hazard Complexity in an Interactive Casual Game of Disaster Preparedness and Response"

_Geoscience Communication, 2020_

## Referee Comment (RC1) · Christopher Skinner (Referee) · 17 Jun 2020

Firstly, apologies for not submitting this reviewers report sooner. This has been a complex review and I wanted to make sure I had properly considered it before submission.

The article covers the development of the Earth Girl Volcano game. The game itself is extremely well researched, with great consideration to every design and game-play element, a high level of attention to detail, and despite the underlying and background complexity, it creates a simple, intuitive gaming experience, that is fun to play over and

over again. This manuscript itself is a reflection of the game and covers in great detail the rationale and design of the game.

The reason the review has been difficult is because the manuscript does not fall within the defined scope of Geoscience Communication. It has been submitted as a research article but there is no research question, methodology, and no form of evaluation to assess that the question has been answered. There is a lot of detail about the game and its design but the authors do not address and evidence whether those design choices were successful in achieving the objectives. Under the journal's manuscript descriptions, this manuscript is simply reporting on a public engagement activity. It doesn't fall in the scope of the other manuscript types either.

"Research articles report substantial new results and conclusions from scientific investigations of geoscience communication initiatives within the scope of the journal. Please note that the journal scope is focused on qualitative and quantitative studies of geoscience communication rather than simply reporting on public engagement initiatives."

https://www.geoscience-communication.net/about/manuscript_types.html

However, as someone who conducts research in this area and is the target audience for Geoscience Communication, I found the manuscript to be immensely insightful and useful, and wherever it is published, either here or elsewhere (in current form, a book chapter would be appropriate), it is work I anticipate I would reference frequently and use as a case study in teaching. The inter-disciplinary backgrounds of the authors provides new perspectives on designing and using games for science and hazard communication that will be valuable for the readership. Ultimately, this is an editorial decision as it is out of scope, but based on the merits of the work, I recommend it for publication with minor corrections.

Key Corrections

The main correction is that some points made need greater evidencing, either through citation or with evidence from evaluations of the game. An example is in the abstract, lines 26-29 - "The combination of all these techniques yields a whole that is greater than the sum of its parts, a perfect storm that is able to create an emotional connection between players and the hazard scenarios in the game." This is an ambition of the game design, and many of the game design elements used have been adopted as previous studies have shown they are effective for engendering emotional connections, yet the authors have not provided the evidence that this game has achieved this, and also that the combined effect is greater than the individual elements. Statements like this need either removing or rewording in a way that states this is the aim and not a finding. I have highlighting further examples in the line-by-line comments below.

The authors need to consider the use of the phrase 'natural disaster' as it is often not helpful in communicating hazard risk and many argue is factually incorrect. The hazard, in this case emerging from volcanic activity, is natural but it only becomes a disaster because of human decisions. There are places where the game is described as a disaster simulation, or that the players manage the disaster, however, I would suggest the game itself is a hazard simulation and the outcome is a disaster or not because of the player's choices. It would be useful to more clearly define these terms and their use throughout.

Editor - stylistically, do game titles need to be italicised?

Line-by-line comments and technical suggestions

Line 15 - Add a comma after 'Unfortunately'

Lines 26-28 - the manuscript does not provide evidence to support this statement and should be removed or reworded

Line 31 - Add 'are' inbetween 'and' and 'exposed'

Lines 46-51 - This paragraph repeats what is said in the opening paragraph. Consider

merging them.

Line 46 - hyphenate easy-to-play and easy-to-learn to be consistent with line 32

Line 48 - A reference is needed for the ranking here. I think it is the same reference used in line 51.

Line 86 - Be careful with use of the term 'natural disaster'. Are disasters natural or because of human actions?

Line 92 - Somewhere in the introduction it would be useful to have a short review of previous hazard awareness games, particularly video games, and place Earth Girl Volcano in context with them. In particular, the video game of Mani et al (2016) has a similar objective but takes a very different approach.https://www.nat-hazards-earth-syst-sci.net/16/1673/2016/

Line 99 - natural disaster again if you wish to change it

Line 110 - volcanic disaster I would suggest is similar to natural disaster

Line 135-136 - Is this rating suggesting the game is well designed to engage this target audience or that it doesn't contain elements that might be inappropriate/harmful for some age groups? The sentence may be misleading due to the sentence immediately before.

Lines 138-139 - I think there is a missing word(s) after 'stay away'

Line 143 - Change 'in' to 'on'

Lines 146-147 - How relevant is this considering Adobe will be removing all support for Flash at the end of 2020? Are there plans to update to html5 or hosting on itch for example?

Lines 148-152 - This is a repeat of information provided in the Introduction.

Line 154 - It would be useful to define what is meant by 'a casual game' for readers

who may not have any gaming background.

Line 159 - Change 'device' to 'devise'

Line 160 - Define what is meant by 'adventure' and 'management' games

Section 1.7, lines 212-231 - This section is really interesting and these insights are valuable. Were these responses recorded in any way that could be used as evidence for this? If not, you need to make it clear that this is anecdotal information from the process and that it is not based on a formal evaluation. Have there been previous studies analysing collaborations between scientists and artists that would support the observations?

Line 224 - My understanding of gamification is that it is the use of gaming elements as part of something but stopping short of making an actual game itself, so the use of the term here wouldn't be correct.

Line 239 - 240 - This statement needs evidencing or removing.

Line 241 - 'classical theatre' has a specific disciplinary meaning (i.e., the theatre of ancient Greece) that could be ambiguous here. 'Traditional theatre' might be more appropriate.

Line 241 - Theatre usually doesn't have a screen therefore don't have screenplays - script or text would be a more suitable term.

Line 250 - 253 - These statements could use evidencing by referencing appropriate sources.

Line 291 - 292 - This sentence needs evidencing or re-wording, for example replacing 'helps' with 'intends'

Line 295 - Suggest replacing 'disaster' with 'hazard' in both cases

Line 316 - How was this observed/recorded? If it is anecdotal, please state that it is

Line 336 - Change 'payer' to 'player'

Line 392 - 411 - Make it clearer in this section when you refer purely to the in-game effect of the tools and when referring to real-world effects. For the latter, this should be accompanied by supporting evidence (like for Sabo dams in lines 402-403).

Line 412 - Add a comma after 'Initially'

Line 418 - Change 'loose' to 'lose'

Line 419 - As a matter of interest, have you tried using a more advanced version with all the tools in to run with practitioners?

Line 426 - Add a comma after 'speaking'

Line 429 - the tool names are capitalised here but not elsewhere. It doesn't matter if there are or are not capitalised but should be consistent.

Line 442 - Change 'cannot mitigated' with 'cannot be mitigated for'

Line 451 - Add 'the' between 'than' and 'others'

Line 458 - Move '(also known as check dams)' to the first reference of sabo dams in line 402.

Line 479 - 480 - It is not enough to say it is well documented, please provide references to this documentation as evidence.

Line 481 - 482 - I'm not sure I understand this sentence - does this refer to people who have evacuated from nearby towns trying to find shelter in the town the player is trying to help?

Line 507 - Remove 'for no good reason' as this could read as they refuse to evacuate until given a good reason to do so.

Line 514 - Change 'he or she' with 'they'

Line 532 - Remove 'interactive game'. It is superfluous here and could be read to mean a different version of the game.

Lines 538 - 548 - Do you have formal observations/evaluations for these? If so, please provide information on the methodology used. If not, please make it clear these are anecdotal observations.

Line 542 - Change 'student' to 'students'

Line 554 - 555 - Are there studies that show this that can support your belief?

Line 578 - Add a comma after 'hazards'

Line 578 - Change 'in which' to 'that'

Line 604 - 609 - All these statements require supporting evidence - how do we know explaining alone does not create emotional response?

Line 616 - 618 - Was this observed via a formal methodology or is anecdotal information?

Line 629 - I know it seems obvious, but you need to provide evidence that suspense and tension create an emotional connection.

Conclusions, lines 647 - 665 - The conclusion is weak and doesn't really say much, but as the manuscript is a detailed description of the design process rather than addressing a research question, this is to be expected. There is reference to user feedback throughout this section - please provide more details of how this was collected and recorded. Has this been a formal evaluation or is it anecdotal?

Line 660 - 661 - It maybe that this is required for publication in Geoscience Communication. If it is decided it is and you wish to continue, I would suggest a focus on how well the game manages to create emotional connections for players, and then how well that leads to increased hazard preparedness.

The manuscript is missing sections -

Data/Code Availability - please include in this section where people can access the game (this information is already in the manuscript but should be repeated here. If any of the information was based on formal evaluation, details of the availability of that data should be placed here.

Author contributions

Conflict of interest statement

It is not listed on the journal website but should there also be a research ethics statement. Thank you for the manuscript, I enjoyed reading it

Chris Skinner

---

## Author Comment (AC1) · 4 Jul 2020

Dear Chris,

Thank you for taking the time to review the paper, for your thoughtful comments, and for the positive comments on our practice-based research. Some of your statements, however, seem narrow-minded in the context of art-science interdisciplinary collaboration, and we reply with a different point of view regarding a Special Issue on Art and Science.

[Figure]

You raise two major issues (natural disasters and the nature of research) and multiple smaller line-by-line comments. Our responses to the latter are summarized at the end of the document. We have amended the spelling mistakes and clarified statements that needed clarification.

NATURAL DISASTERS

REVIEWER: The authors need to consider the use of the phrase 'natural disaster' as it is often not helpful in communicating hazard risk and many argue is factually incorrect. The hazard, in this case emerging from volcanic activity, is natural but it only becomes a disaster because of human decisions.

AUTHORS' RESPONSE: We have amended "natural disasters" to "natural hazards." We agree that in the context of Earth science natural hazards generally turn into disasters because of human inaction or lack of preparedness. One could argue other points of view but this paper is not the forum for that discussion.

RESEARCH COMMENT

REVIEWER: The reason the review has been difficult is because the manuscript does not fall within the defined scope of Geoscience Communication. It has been submitted as a research article but there is no research question, methodology, and no form of evaluation to assess that the question has been answered. There is a lot of detail about the game and its design but the authors do not address and evidence whether those design choices were successful in achieving the objectives. Under the journal's manuscript descriptions, this manuscript is simply reporting on a public engagement activity. It doesn't fall in the scope of the other manuscript types either.

AUTHORS' RESPONSE: We understand that SCIENTIFIC RESEARCH is based on a specific method that is centered on quantitative data. The parameters of modern scientific research are narrowly defined and have been widely used for over a century. PRACTICE-BASED RESEARCH is another form of research that also generates

Interactive
comment
**[GCD](GCD)**

Interactive
comment

knowledge, and this type of research started to be formulated and formalized at the turn of the 21st century. The later is different from scientific research and it is the de-facto research methodology used in the creative arts. You can learn more details about this approach in this MIT Journals free-access paper: "Practice-Based Research in the Creative Arts: Foundations and Futures from the Front Line" by Linda Candy and Ernest Edmonds. https://doi.org/10.1162/LEON_a_01471

Clearly we failed to bring up in the manuscript the issue of practice-based research. We are now incorporating a few extra lines about that it in the abstract and the conclusions.

The research behind the Earth Girl Volcano game is practice-based research, and the aim of the paper is to document the new knowledge regarding a methodology for creating games. We believe that our unique and original methodology and approach to creating games can have a deep impact in the creation of science-inspired games that people and non-scientists want to play. Designing and producing earth science GAMES that are engaging, based on science, and can help communities to increase understanding and preparedness is a lot more than "simply reporting on a public engagement activity."

We believe that our manuscript belongs in this Special Issue of EGU's Journal of Geoscience Communication because it presents an innovative game design and game methodology that we crafted (i.e. developed, invented) through our practice-based research. Narrowly defined the Earth Girl Volcano game is not scientific research. The aim of our research and manuscript is not to measure the game and/or to provide quantitative information about it. We value quantitative information but that is not what this research/manuscript is about.

Our aim in this paper is to reveal our design methodology. As stated by the reviewer, this game "is extremely well researched, with great consideration to every design and game-play element, a high level of attention to detail, and despite the underlying and background complexity, it creates a simple, intuitive gaming experience, that is fun to
play over and over again." All that was precisely the aim and the result of our research. Our "simple, intuitive gaming experience, that is fun to play over and over again" was the result of methodical development that placed the player, not the quantitative or technical information, at the center of the process. The paper documents and reflects on that approach.

Academic discussions about the nature of research aside, our goal as artists and scientists is to advance the usefulness of Earth science to average people. It seems narrow-minded to expect artists to behave like scientists, to talk like scientists and to think like scientists. It seems counter-productive to judge the output of art-and-science collaborations merely by the rigid metrics of traditional science. What would then be the point of collaborations between artists and scientists? The whole point of interdisciplinary collaboration is to create a space where experiments can happen, where we cant try new ways to create, to think about our process, and to evaluate our results.

"Thinking outside the box" is a bit of a cliché but the Earth Girl Volcano game would not exist if we had followed the traditional "square" scientific quantitative approach to crafting "serious" educational games. Hence the value of our paper in the context of practice-based research and interdisciplinary collaboration.

To conclude, we believe that this article can greatly enhance the value of this Special Issue, and that the work presented is an encouraging example of the innovative results of an exemplary collaboration between artists and scientists in the context of Earth science and hazard preparedness.

LINE-BY-LINE COMMENTS

Editor - stylistically, do game titles need to be italicised? - YES, CLARIFIED by Copernicus Publications, DONE

Line 15 - Add a comma after 'Unfortunately' - DONE

Lines 26-28 - the manuscript does not provide evidence to support this statement and

should be removed or reworded - FIXED

Line 31 - Add 'are' inbetween 'and' and 'exposed' - DONE

Lines 46-51 - This paragraph repeats what is said in the opening paragraph. Consider merging them. - KEPT AS IS, we like the slight repetition as it reinforces key concepts

Line 46 - hyphenate easy-to-play and easy-to-learn to be consistent with line 32 - DONE

Line 48 - A reference is needed for the ranking here. I think it is the same reference used in line 51. - KEPT, it is the same reference.

Line 86 - Be careful with use of the term 'natural disaster'. Are disasters natural or because of human actions? - DONE

Line 92 - Somewhere in the introduction it would be useful to have a short review of previous hazard awareness games, particularly video games, and place Earth Girl Volcano in context with them. In particular, the video game of Mani et al (2016) has a similar objective but takes a very different approach. https://www.nat-hazards-earthsyst-sci.net/16/1673/2016/ - ADDED a short sentence, but additional extensive information is beyond the scope of the paper

Line 99 - natural disaster again if you wish to change it - DONE

Line 110 - volcanic disaster I would suggest is similar to natural disaster - AGREE

Line 135-136 - Is this rating suggesting the game is well designed to engage this target audience or that it doesn't contain elements that might be inappropriate/harmful for some age groups? The sentence may be misleading due to the sentence immediately before. BOTH, just a statement of fact.

Lines 138-139 - I think there is a missing word(s) after 'stay away' - DONE

Line 143 - Change 'in' to 'on' - DONE

Lines 146-147 - How relevant is this considering Adobe will be removing all support for Flash at the end of 2020? Are there plans to update to html5 or hosting on itch for example? - DONE, it is a factual statement of present availability, added possible alternatives.

Lines 148-152 - This is a repeat of information provided in the Introduction. - AGREE, but we like it as is for the sake of clarity.

Line 154 - It would be useful to define what is meant by 'a casual game' for readers who may not have any gaming background. - CASUAL GAMES are casually defined throughout the paragraph.

Line 159 - Change 'device' to 'devise' - DONE

Line 160 - Define what is meant by 'adventure' and 'management' games - ALREADY list the key characteristic of each, more details beyond scope of paper.

Lines 212-231, Section 1.7 - This section is really interesting and these insights are valuable. Were these responses recorded in any way that could be used as evidence for this? If not, you need to make it clear that this is anecdotal information from the process and that it is not based on a formal evaluation. Have there been previous studies analysing collaborations between scientists and artists that would support the observations? - LEFT AS IS, thank you, left as is. These are meant as simple side thoughts about the process of interdisciplinary collaboration, to quickly illustrate our general approach. It is not meant to be an exhaustive exploration of interdisciplinary collaboration between artists and scientists. That would be a different paper.

Line 224 - My understanding of gamification is that it is the use of gaming elements as part of something but stopping short of making an actual game itself, so the use of the term here wouldn't be correct. - GAMIFICATION is a general open-ended term, my understanding of gamification includes making actual games, we'd like to keep it inclusive and simple.
Line 239 - 240 - This statement needs evidencing or removing. - MODIFIED

Line 241 - 'classical theatre' has a specific disciplinary meaning (i.e., the theatre of ancient Greece) that could be ambiguous here. 'Traditional theatre' might be more appropriate. - FIXED, simplified

Line 241 - Theatre usually doesn't have a screen therefore don't have screenplays - script or text would be a more suitable term. - DONE

Line 250 - 253 - These statements could use evidencing by referencing appropriate sources. - BEYOND SCOPE, meant in an anecdotal way. Also these are concept of introductory courses to Storytelling and we want to stay away from explaining and referencing basic concepts of dramatic writing and film production. We'd like to keep the manuscript as focused on the game as possible, and short too.

Line 291 - 292 - This sentence needs evidencing or re-wording, for example replacing 'helps' with 'intends' - DONE

Line 295 - Suggest replacing 'disaster' with 'hazard' in both cases - DONE

Line 316 - How was this observed/recorded? If it is anecdotal, please state that it is - CLARIFIED, methodical

Line 336 - Change 'payer' to 'player' - DONE

Line 392 - 411 - Make it clearer in this section when you refer purely to the in-game effect of the tools and when referring to real-world effects. For the latter, this should be accompanied by supporting evidence (like for Sabo dams in lines 402-403). - FIXED, ADDED info on crossing rivers and community education

Line 412 - Add a comma after 'Initially' - DONE

Line 418 - Change 'loose' to 'lose' - DONE

Line 419 - As a matter of interest, have you tried using a more advanced version with

all the tools in to run with practitioners? - YES, FYI ONLY [not in manuscript], early in the process we had a build of the game with an extended toolset, but this overwhelmed the large majority of the test-players and we didn't pursue that avenue. We didn't have the resources to build two versions in parallel so we stuck with the limited but easy-to-handle toolset.

Line 426 - Add a comma after 'speaking' - DONE

Line 429 - the tool names are capitalised here but not elsewhere. It doesn't matter if there are or are not capitalised but should be consistent. - FIXED

Line 442 - Change 'cannot mitigated' with 'cannot be mitigated for' - DONE

Line 451 - Add 'the' between 'than' and 'others' - DONE

Line 458 - Move '(also known as check dams)' to the first reference of sabo dams in line 402. - DONE

Line 479 - 480 - It is not enough to say it is well documented, please provide references to this documentation as evidence. - ADDED reference from Al Jazeera News [refuse to evacuate]

Line 481 - 482 - I'm not sure I understand this sentence - does this refer to people who have evacuated from nearby towns trying to find shelter in the town the player is trying to help? - CORRECT, rephrased, hope it is more clear

Line 507 - Remove 'for no good reason' as this could read as they refuse to evacuate until given a good reason to do so. - DONE

Line 514 - Change 'he or she' with 'they - DONE

Line 532 - Remove 'interactive game'. It is superfluous here and could be read to mean a different version of the game. - LEFT AS IS, for the sake of clarity

Lines 538 - 548 - Do you have formal observations/evaluations for these? If so, please

Interactive
comment

provide information on the methodology used. If not, please make it clear these are anecdotal observations. - FIXED, added information

Line 542 - Change 'student' to 'students' - DONE

Line 554 - 555 - Are there studies that show this that can support your belief? [emotional connection] - YES but beyond the scope of paper, we prefer to keep it simple with the use of "believe"

Line 578 - Add a comma after 'hazards' - DONE

Line 578 - Change 'in which' to 'that' - DONE

Line 604 - 609 - All these statements require supporting evidence - how do we know explaining alone does not create emotional response? - EXPANDED, these are our guiding principles, this approach is what helps us to produce what we create, to substantiate the psychology of emotion and learning is beyond the scope of this paper

Line 616 - 618 - Was this observed via a formal methodology or is anecdotal information? - OBSERVED, CLARIFIED in manuscript

Line 629 - I know it seems obvious, but you need to provide evidence that suspense and tension create an emotional connection. - LEFT AS IS, this is Dramatic Writing 101, want to stay away from very basic concepts

Lines 647 - 665, Conclusion - The conclusion is weak and doesn't really say much, but as the manuscript is a detailed description of the design process rather than addressing a research question, this is to be expected. There is reference to user feedback throughout this section - please provide more details of how this was collected and recorded. Has this been a formal evaluation or is it anecdotal? - IMPROVED, CLARIFIED

Line 660 - 661 - It maybe that this is required for publication in Geoscience Communication. If it is decided it is and you wish to continue, I would suggest a focus on how well

the game manages to create emotional connections for players, and then how well that leads to increased hazard preparedness. - FUTURE WORK, beyond scope of current manuscript

Data/Code Availability - please include in this section where people can access the game (this information is already in the manuscript but should be repeated here. If any of the information was based on formal evaluation, details of the availability of that data should be placed here. - ADDED section Game Availability At time of writing the public can download the English computer version of the Earth Girl Volcano game for Windows and MacOS from https://earthobservatory.sg/ and https://art-science-media.com/. Several localizations of the game for digital tablets are also available at Google Play and the App Store. The early Earth Girl Flash games are available at http://earthgirlgame.com/ and they require the Flash plug-in. Before the end of 2020 we plan to upload the Flash games to the repository website https://itch.io/.

Author contributions - ADDED section Isaac Kerlow conceptualized and designed the Earth Girl Volcano game, supervised the development and production team throughout the process, and wrote the original draft of the published work. Gabriela Pedreros and Helena Albert contributed to the methodology, participated in the validation, and reviewed and edited the published work.

Conflict of interest statement - ADDED section Competing Interests The authors declare that they have no conflict of interest.

ALSO ADDED Video Supplements Two short videos illustrate the features of this game. The Earth Girl Volcano - Game Overview video (1m 30s) provides an introduction to this casual strategy game. Available at https://vimeo.com/294752004 and also at DOI (pending). The Earth Girl Volcano - Game Demo video (6m 27s) illustrates a sequential gameplay of the game. Available at https://vimeo.com/305156140 and also at DOI (pending).

---

## Referee Comment (RC2) · Anonymous Referee #2 · 15 Jul 2020

This manuscript focuses on educating and engaging the general public of volcanic hazards in relevant communities, and thus falls squarely within the scope of this journal, which includes "geoscience engagement".

The game environment has for some time already been a very successful means employed by non-government entities and government agencies alike, to educate and engage the public. This particular game aims to communicate and personalize, even empathize the experience of volcanic hazards and behavioral responses in a culturally sensitive context. As such, the impact and demonstrated response of the public trials

of this game are a significant and substantial contribution to the field of communication of, and public engagement in science.

The technical quality of the work described is of a high standard, and the authors convincingly and quantitatively demonstrate their diligence in providing a culturally, scientifically, and artistically well-balanced product. Furthermore, their detailed narrative of methods and lessons-learned are very valuable for any other disaster-related public engagement and communication game-environment endeavor.

To the best of our knowledge, it is the first of its kind (among natural hazards game engagement tools of communication) to undergo such careful, diligent, and methodical rigorous vetting and designing to provide a high quality product.

The technical approach and applied methods are excellent. However, the discussion and written presentation/delivery of the results lack balance and structure.

The presentation quality is not sufficient for publication just yet. I suggest revisions below, which while changing the structure and flow of the manuscript, will hopefully improve its delivery significantly. With improvements (revision), it would make a possibly quite impactful and valuable article in Geoscience Communication.

**Detailed comments and suggestions:**

The authors should take care to adjust any possibly marketing-like language, which is inappropriate, and replace it with factual language. A publication in this journal should not so much evoke emotions, which is the domain of marketing, but convincingly document the scope, methods, results, and lessons learned.

Otherwise, the language used makes the paper read fluently. It appears to be very long, though I didn't check the word count vs. the journal's recommendations.

The abstract may benefit from framing it more in a communication theory context, which could be easily solved with careful wording choices and better structuring.

The manuscript structure could be significantly improved by better header hierarchy to provide a more structured and easier to follow logic, and by providing brief introductions. For instance, It is unclear where the context introduction ends and the methods section begin.

I suggest to add a table of user testing statistics, if available, since this is an important foundation to the resilience and impact of the final product. It could also be a pie chart, bar chart, or similar, and should be referred to in the text at the appropriate sections.

Section 2 will need a paragraph or so of introduction of the following sections. I suggest to number start with a preamble providing a brief description of the sections that follow (no more than one sentence or bullet per section) and call it #2. Call "Act one" #2.1 (levels of difficult, #2.1.1); call "Act 2" 2.2, and so forth. Of course, if my earlier comment on methods is taken into account, these all change to 3, 3.1, 3.2, etc.

Section 5 is focused on behavioral and psychological methods to communicate functional relationships and provide a learning experience feedback component. I suggest to expand on the preamble of this section in a communication theory introduction of game engagement and education state-of-the-art knowledge from the relevant literature. The following subsections 5.1 through 5.3 could be listed as bullets in this preamble and given appropriate context.

Section 5.1 "Rules of Thumb" as a title phrase is based on a European/American language-centric idiom, and I suggest to use different language (or define "rules of thumb" in the beginning). There is more technical language that could be used to describe these relationships described in Sect. 5.1, like process feedbacks, functional relationship, etc., per their technical definitions. The author should recall that the audience here is not the game audience.

Section 5.4 - it is not obvious to me why this shouldn't be part of the methods section in the beginning.

The Conclusions appear to lack context from the game engagement/education communication literature, which I am not familiar with. It also lacks a communication-focused impacts summary (how does this science engagement/education game impact society and behavior, or how is it projected to do so?).

I suggest the author look at a few examples of successful publications from this journal, too, to possibly improve the delivery in terms of length, level of detail, structure, and framing.
* * *

---

## Author Comment (AC2) · 29 Aug 2020

Dear Reviewer #2,

Thank you for taking the time to read the manuscript and provide feedback.

We have numbered your comments and our replies for ease of tracking.

Thank you again,

The authors

[Figure]

REVIEWER 1 - The authors should take care to adjust any possibly marketing-like language, which is inappropriate, and replace it with factual language. A publication in this journal should not so much evoke emotions, which is the domain of marketing, but convincingly document the scope, methods, results, and lessons learned.

AUTHORS 1 - The manuscript documents interdisciplinary work that involves art and science. Emotions are the basis of storytelling and artistic creation, which are at the core of this game. The game incorporates notions ofÂă Earth science and evacuation management, but the game itself is not a scientific project nor an emergency evacuation manual. The game is the result of interdisciplinary artistic and scientific collaboration and because of that we use language that is used to describe artistic methods and techniques. Marketing on the other hand is oftentimes based on partial truths that may create a false impression of things, there is no intention to mislead or anything misleading about this game. We will remove any language that may sound like marketing. Any specific recommendations would beÂăappreciated.

REVIEWER 2 - Otherwise, the language used makes the paper read fluently. It appears to be very long, though I didn't check the word count vs. the journal's recommendations.

AUTHORS 2 - At 22 pages the manuscript is a few pages beyond the 15-pp. MINIMUM length specified by the special issue guidelines.

REVIEWER 3 - The abstract may benefit from framing it more in a communication theory context, which could be easily solved with careful wording choices and better structuring.

AUTHORS 3 -ÂăWe choose to focus the abstract not only on the storytelling method, but also on key aspects about game design and implementation of preparedness and response concepts in the game mechanics. Following the reviewer suggestion we can include some specific terminology to make more clear the importance of "storytelling" as a powerful communication method.
REVIEWER 4 - ĂăThe manuscript structure could be significantly improved by better header hierarchy to provide a more structured and easier to follow logic, and by providing brief introductions. For instance, it is unclear where the context introduction ends and the methods section begin.

AUTHORS 4 - We believe that the manuscript follows the header nomenclature suggested by the journal. Section 1, Introduction, includes introductory notes on both context and methodĂă(divided in sections 1.1 to 1.7). The subsequent sections focus primarily on the different methods used in each of the story acts (one manuscript section per story act). As a possibility we could group the story acts under one single section and include a brief introductory paragraph as suggested by the reviewer in comment 6.

REVIEWER 5 - ĂăI suggest to add a table of user testing statistics, if available, since this is an important foundation to the resilience and impact of the final product. It could also be a pie chart, bar chart, or similar, and should be referred to in the text at the appropriate sections.

AUTHORS 5 – Even though quantitative data would add value to the manuscript,Ăădetailed testing statistics are beyond the scope of this specific manuscript.

REVIEWERĂăÂă6- Section 2 will need a paragraph or so of introduction of the following sections. I suggest to number start with a preamble providing a brief description of the sections that follow (no more than one sentence or bullet per section) and call it #2. Call "Act one" #2.1 (levels of difficult, #2.1.1); call "Act 2" 2.2, and so forth. Of course, if my earlier comment on methods is taken into account, these all change to 3, 3.1, 3.2, etc.

AUTHORS 6 - It is unclear why further subsection introductions would be necessary within current Section 2 since the section is short and topics are already contextualized. We prefer to have a separate section for each Act in the story as each one of them

follows a unique approach, quite different from the others. Consolidating all the acts in a single section would dilute the clarity of the story analysis. Each act functions as a different moment in the overall process of preparedness and response, and we mean to make this obvious by analyzing each of the acts in a separate manuscript section.

REVIEWER 7 - Section 5 is focused on behavioral and psychological methods to communicate functional relationships and provide a learning experience feedback component. I suggest to expand on the preamble of this section in a communication theory introduction of game engagement and education state-of-the-art knowledge from the relevant literature. The following subsections 5.1 through 5.3 could be listed as bullets in this preamble and given appropriate context.

AUTHORS 7 - Section 5 is focused on storytelling, game design and game mechanic techniques used to characterize situations of preparedness and response. Further expansion on additional behavioral and psychological issues is beyond the scope of this paper.

REVIEWER 8 - Section 5.1 "Rules of Thumb" as a title phrase is based on a European/American language-centric idiom, and I suggest to use different language (or define "rules of thumb" in the beginning). There is more technical language that could be used to describe these relationships described in Sect. 5.1, like process feedbacks, functional relationship, etc., per their technical definitions. The author should recall that the audience here is not the game audience.

AUTHORS 8 - We will consider more precise terms to describe the ideas presented in subsection 5.1. We will find a better way to address "Rules of Thumb"? We realize that the readers of the paper will not be primarily the game audience but we recognize a great value in exposing Earth scientists to effective game design methods and game terminology. Familiarity with game terminology and methodology may help Earth scientists to become more effective collaborators in interdisciplinary projects that involve gaming. Because this manuscript was submitted to an interdisciplinary special issue

we expect a wide variety of readers, including artists who are looking for reasons and encouragement to collaborate with Earth scientists.

REVIEWER 9 - Section 5.4 - it is not obvious to me why this shouldn't be part of the methods section in the beginning.

AUTHORS 9 - Simulation and scripting are two opposing gaming techniques rarely used together in a science-inspired game to yield a convincing gameplay experience. We use both simulation and scripting techniques to characterize volcanic hazard complexity, and Section 5 consolidates all the characterization techniques used in the game. For added clarity we present this section after the structure analysis of the three different acts in the game.

REVIEWER 10 - The Conclusions appear to lack context from the game engagement/education communication literature, which I am not familiar with. It also lacks a communication-focused impacts summary (how does this science engagement/education game impact society and behavior, or how is it projected to do so?).

AUTHORS 10 - We will develop the impacts, we will improve the Conclusion in the context of game-design and game process (how does this game innovate in terms of characterizing natural hazards? What lessons were learned to improve an interdisciplinary collaboration?)

---

## Author Comment (AC3) · 9 Sep 2020

Dear Jutta,

We posted our response to review #2 about a week ago.

Please let us know if you have any further questions or suggestions.

Thank you and regards,

Isaac and team